# MULTI-AGENT BAYESIAN OPTIMIZATION WITH COUPLED BLACK-BOX AND AFFINE CONSTRAINTS

## ABSTRACT

This paper studies the problem of distributed multi-agent Bayesian optimization with both coupled black-box constraints and known affine constraints. A primal-dual distributed algorithm is proposed that achieves similar regret/violation bounds as those in the single-agent case for the black-box objective and constraint functions. Additionally, the algorithm guarantees an $\mathcal{O}(N\sqrt{T})$ bound on the cumulative violation for the known affine constraints, where $N$ is the number of agents. Hence, it is ensured that the average of the samples satisfies the affine constraints up to the error $\mathcal{O}(N/\sqrt{T})$. Furthermore, we characterize certain conditions under which our algorithm can bound a stronger metric of cumulative violation and provide best-iterate convergence without affine constraint. The method is then applied to both sampled instances from Gaussian processes and a real-world optimal power allocation problem for wireless communication; the results show that our method simultaneously provides close-to-optimal performance and maintains minor violations on average, corroborating our theoretical analysis.

## 1 INTRODUCTION

Bayesian optimization (BO), as a sample-efficient black-box optimization method (Frazier, 2018), has found wide application in tuning hyperparameters of machine learning models (Snoek et al., 2012), discovering new drugs (Negoescu et al., 2011), and optimizing the performance of energy systems (Xu et al., 2023b), etc.. It is particularly useful when the objective function is expensive to evaluate and potentially multi-modal.

Bayesian optimization is based on surrogate modeling of the unknown black-box objective function. Specifically, the black-box function is assumed to be sampled from a Gaussian process. The Gaussian process posterior is updated as a new function evaluation is obtained. To decide the next sample point, an acquisition function, such as expected improvement (Jones et al., 1998), or upper confidence bound (Srinivas et al., 2012), is optimized. One then samples the optimizer of the acquisition function in the hope of identifying the global optimum within as few samples as possible.

One challenge of Bayesian optimization is the existence of black-box constraints present in many physical systems. For example, when tuning the parameters of a chemical reactor, one needs to keep the residue fractions of some chemical components below predefined thresholds while maximizing the economic profit (del Rio Chanona et al., 2021). Many algorithms have been proposed to deal with constraints, including CEI (Gardner et al., 2014; Gelbart et al., 2014), SafeOPT (Sui et al., 2015), ADMMBO (Ariafar et al., 2019), penalty methods (Xu et al., 2022b; Lu & Paulson, 2022; Guo et al., 2023), primal-dual method (Zhou & Ji, 2022) and the recent CONFIG (Xu et al., 2023a).

Despite the popularity and success of (constrained) Bayesian optimization in numerous science and engineering applications (Shahriari et al., 2015), the current development of BO mostly focuses on the case of one single agent. However, many real-world black-box optimization problems involve *multiple agents*. The objective and constraints of those agents can be coupled in an *additive* way. For example, for some demand response formulations (Vardakas et al., 2014) in a smart grid, multiple consumers adapt their local electricity consumption habits to maximize their individual utilities while a global total energy consumption constraint over those consumers is imposed.

Compared to the conventional single-agent scenario, the multi-agent setting introduces several new challenges. First, the black-box function evaluations need to be done locally. In practice, these

evaluations may correspond to real-world physical experiments with local facilities. For example, in building control for demand response (Chen et al., 2018), black-box function evaluations correspond to measuring the occupants' utilities (e.g., thermal comfort) and energy consumption in a building. Due to privacy issues or limited communication bandwidth, the agents may not want to share the exact local evaluation data with other agents. Secondly, the acquisition step needs to be distributed. Agnostic application of the conventional Bayesian optimization method in a centralized way may suffer from a severe *curse of dimensionality*, since the number of agents can be large. Thirdly, there may be known affine constraints, which capture the consensus or coordination among the agents, in addition to the black-box constraints in BO (Gelbart et al., 2014; Gardner et al., 2014). For example, when tuning the optimal speed for vehicle platooning (Xu et al., 2022a), all the vehicles' speeds need to be the same. In another example of power allocation for wireless communication, the summation of allocated power needs to be equal to a total power budget (Tse, 1997).

Existing works on multi-agent Bayesian optimization are mostly heuristic. An ADMM-based multi-agent Bayesian optimization algorithm is proposed in (Krishnamoorthy & Paulson, 2023) without any guarantees on regret or violations. In addition, there are also existing works that only consider a single objective but distribute the black-box function evaluations over multiple agents (Wu & Frazier, 2016; Kandasamy et al., 2018; Daulton et al., 2021; Ma et al., 2023). Additive structure is also exploited to boost the sample efficiency of Bayesian optimization (Kandasamy et al., 2015; Gardner et al., 2017; Rolland et al., 2018). Another line of works on federated Bayesian optimization (Dai et al., 2020; 2021) and federated kernelized bandits (Li et al., 2022; Salgia et al., 2022) consider the setting where a group of agents aim to accelerate their local black-box optimization algorithms by leveraging the information from other agents. However, these three lines of research do not consider coupled constraints caused by multiple agents. In addition to the literature on Bayesian optimization, the general problem of distributed optimization in multi-agent systems has also gained wide interest. The readers are referred to the surveys (Nedić & Liu, 2018; Yang et al., 2019) and references therein. The works most relevant to this paper are on zero-order distributed non-convex optimization (Tang et al., 2020). However, these gradient estimation based methods can only guarantee convergence to a local optimum and may suffer from severe regret as compared to the global optimum. In contrast, we aim to develop a distributed algorithm with certain global optimality properties in this paper.

This paper proposes a distributed multi-agent Bayesian optimization algorithm with both additive coupled black-box and known affine constraints. Specifically, our contributions include:

- We propose a primal-dual distributed algorithm to solve the multi-agent Bayesian optimization problem with additive objective/constraints. Our algorithm achieves similar regret and violation (of black-box constraint) bounds as those in the single-agent case (Zhou & Ji, 2022), up to a multiplicative term depending on the number of agents. As far as we know, our algorithm is the *first* distributed multi-agent BO algorithm that enjoys theoretical regret/violation bounds.

- In addition, the cumulative violation of the affine constraints can be upper bounded by $\mathcal{O}(N\sqrt{T})$, where $N$ is the number of agents and $T$ is the running horizon length.

- Furthermore, we characterize certain conditions under which our algorithm can provide sublinear bounds on cumulative strong violation (accumulation of the violated part) for the black-box constraint and best-iterate convergence.

- We conduct numerical experiments on both sampled instances from the Gaussian process and a real-world optimal power allocation problem. The results corroborate our theoretical analysis.

Essentially, we leverage the recent constrained kernelized multi-armed bandits algorithm (Zhou & Ji, 2022) to develop a distributed algorithm for multi-agent Bayesian optimization. As compared to (Zhou & Ji, 2022), we introduce additional known coupled affine constraints, which is common in the multi-agent setting. This brings a new coordination challenge in addition to the regret/violation tradeoff and requires a new set of analysis techniques. Furthermore, the conditional bounds on strong violations and best-iterate convergence complement the empirical observations that the primal-dual method can also achieve good performance with respect to these stronger metrics (Zhou & Ji, 2022).

## 2 PROBLEM FORMULATION

We consider a set of agents $[N] := \{1, 2, \cdots, N\}$. Each agent has a local decision variable $x_i \in \mathcal{X}^i \subset \mathbb{R}^{n_i}$ and aims to minimize its local black-box objective function $f_i : \mathcal{X}^i \to \mathbb{R}$.

At the same time, the agent $i$ measures the black-box constraint value $g_i(x_i)$ with local decision $x_i$, where $g_i : \mathcal{X}^i \to \mathbb{R}^m$ and $m$ is the number of black-box constraints. The global constraints $\sum_{i=1}^{N} g_i(x_i) \leq 0$ are imposed on the agents. In addition, the agents need to follow a set of affine constraints $\sum_{i=1}^{N} A_i x_i = b$, which captures the consensus or decision coordination constraints (e.g., resource allocation under budget constraint). Our problem can be formulated as,

$$\min_{x_i \in \mathcal{X}^i, i \in [N]} \quad \sum_{i=1}^{N} f_i(x_i), \quad \text{subject to:} \quad \sum_{i=1}^{N} g_i(x_i) \leq 0, \quad \text{and} \quad \sum_{i=1}^{N} A_i x_i = b, \quad (1)$$

where $f_i, g_i, i \in [N]$ are all local black-box functions, the inequality is interpreted elementwise, $A_i \in \mathbb{R}^{l \times n_i}, i \in [N]$ are known matrices, and $b \in \mathbb{R}^l$ is a known vector. The multi-agent black-box optimization problem formulated in Eq. (1) widely appears in many applications, where $g_i(\cdot)$ may represent certain types of resources (subtracting some thresholds) with global constraints. Examples include matching vehicles and passengers in ride-sharing (Lin et al., 2019), resource allocation in cloud computing (Gao et al., 2020), and demand response in a smart grid (Davarzani et al., 2019).

We aim to solve the problem (1) in a *distributed* and *online* fashion. Specifically, in each round $t$, the agent $i$ can only locally decide the variable $x_i^t$ and locally sample the black-box objective function $f_i$ and the constraint function $g_i$ by conducting software simulation or hardware experiment. Then, the agents can communicate useful information following a scheme before deciding on the next local sample point. We aim to jointly design the local acquisition policy and the communication scheme so that the agents cooperatively solve the problem (1) in a distributed and online fashion.

**Remark 1 (Constraint Formulation)** *The black-box constraint in (1) considers the generic form of taking summation over all the agents. The case of summing over a subset of agents (even only one agent) can be covered by setting the other agents' corresponding constraints to zero functions, with all the following algorithm design and theoretical analysis still holding.*

We make some regularity assumptions regarding the elements in problem (1).

**Assumption 1 (Compact Set and Feasibility)** $\forall i \in [N]$, $\mathcal{X}^i$ *is compact. Furthermore, problem (1) is feasible and its optimal solution $x^\star := (x_1^\star, \cdots, x_N^\star)$ exists.*

Assumption 1 is common in practice. For example, we can usually restrict the set $\mathcal{X}^i$ to a hyper-box when tuning the hyperparameters of a machine learning model. Feasibility is a common assumption in the safe or constrained Bayesian optimization literature (Sui et al., 2015; Xu et al., 2023a).

**Assumption 2 (Regularity)** $f_i \in \mathcal{H}_{i,0}, g_{i,j} \in \mathcal{H}_{i,j}, \forall i \in [N], \forall j \in [m]$, *where $g_{i,j}$ is the $j$-th element of $g_i$, $\mathcal{H}_{i,j}, i \in [N], j \in \{0\} \cup [m]$ is a reproducing kernel Hilbert space (RKHS) equipped with the kernel function $k_{i,j}(\cdot, \cdot) : \mathbb{R}^{n_i} \times \mathbb{R}^{n_i} \to \mathbb{R}$ (See (Schölkopf et al., 2001)). Furthermore, $\|f_i\| \leq C_{i,0}, \|g_{i,j}\| \leq C_{i,j}, \forall i \in [N], j \in [m]$, where $\|\cdot\|$ is the norm induced by the inner product of the corresponding RKHS without further notice. Furthermore, we assume there is a uniform upper bound $\bar{C}$ for $C_{i,j}, \forall i \in [N], j \in \{0\} \cup [m]$, which is independent of the number of agents $N$.*

Intuitively, Assumption 2 means that the black-box functions are regular in the sense of having bounded norms in some RKHSs. It means the black-box functions have a certain 'smoothness' property, at least to a certain degree (see (Schölkopf et al., 2001)). Having a bounded norm in an RKHS is a common assumption in existing Bayesian optimization or kernelized multi-armed bandit literature (e.g., (Srinivas et al., 2012; Chowdhury & Gopalan, 2017a; Zhou & Ji, 2022)).

**Assumption 3 (Observation Model)** *Each agent $i$, $i \in [N]$ has access to a noisy zero-order oracle, which means each round of query $x_i^t, i \in [N]$ returns the noisy function evaluations,*

$$y_{i,0}^t = f_i(x_i^t) + \nu_{i,0}^t \;\;, \;\; y_{i,j}^t = g_{i,j}(x_i^t) + \nu_{i,j}^t \;\;, \quad j \in [m] \quad (2)$$

*where $\nu_{i,j}^t, i \in [N], j \in \{0\} \cup [m]$ is independent and identically distributed $\sigma$-sub-Gaussian noise.*

In practice, the zero-order oracle in Assumption 3 may correspond to real-world physical experiments or software simulations, which can only be accessed by each agent locally.

**Notations** Throughout this paper, we use the notation $X_t := (x^1, x^2, \cdots, x^t)$ to define the sequence of sampled points up to step $t$, where $x^\tau := (x_i^\tau)_{i=1}^N$. Therefore, the historical evaluations are $\mathcal{D}_t := \{(x^\tau, y^\tau)\}_{\tau=1}^t$, where $y^\tau := (y_{i,j}^\tau)_{i \in [N], j \in \{0\} \cup [m]}$. We use $x$ to denote the vertical concatenation of $x_i, i \in [N]$, $\mathcal{X}$ to denote $\prod_{i=1}^N \mathcal{X}^i$ and $n$ to denote $\sum_{i=1}^N n_i$. The notations $f(x) := \sum_{i=1}^N f_i(x_i)$, $g(x) := \sum_{i=1}^N g_i(x_i)$, and $C_j := \sum_{i=1}^N C_{i,j}, j \in \{0\} \cup [m]$ are also used. We use $A \in \mathbb{R}^{l \times n}$ to denote $[A_1 \ A_2 \ \cdots \ A_N]$. Hence, the affine constraint can also be written as $Ax = b$. For simplicity, $[\cdot]^+$ is used to represent the function $\max\{0, \cdot\}$. When applied to a vector, $\|\cdot\|$ is by default the Euclidean norm. $\|\cdot\|_p$ is the standard $p$-norm.

**Assumption 4 (Normalized Kernel)** *The kernel functions are all normalized, such that,* $k_{i,j}(x_i, x_i) \le 1, \forall x_i \in \mathcal{X}^i, i \in [N], j \in \{0\} \cup [m]$.

Most commonly used kernel functions (including the squared exponential kernel and the Matérn kernel) can be normalized in a compact set $\mathcal{X}^i$ and thus satisfy this assumption.

**Assumption 5 (Slackness)** *There exists $\xi > 0$ and a joint probability distribution $\bar{\pi}$ supported over $\mathcal{X}$, such that,*

$$\mathbb{E}_{\bar{\pi}}[g(x)] \le -\xi e, \ \ and \ \ \mathbb{E}_{\bar{\pi}}[Ax] = b, \tag{3}$$

*where $e \in \mathbb{R}^m$ is the vector with all 1s and the inequality is interpreted elementwise.*

Assumption 5 is a very mild slackness assumption on the distributions over the compact set $\mathcal{X}$. We further make some regularity assumptions regarding $\mathcal{X}$ and $A$.

**Assumption 6** *The matrix $A$ is full row rank and there exists $\tilde{x}$ and $\tilde{\rho} > 0$, such that $A\tilde{x} = b$ and $\mathcal{B}_{\tilde{\rho}}^n[\tilde{x}] \subset \mathcal{X}$, where $\mathcal{B}_{\tilde{\rho}}^n[\tilde{x}] := \{x \in \mathbb{R}^n | \|x - \tilde{x}\| \le \tilde{\rho}\}$. Furthermore, $\forall x \in \mathcal{B}_{\tilde{\rho}}^n[\tilde{x}], g(x) \le 0$.*

Assumption 6 is also mild. Full row rank assumption is mild since if $A$ is not full row rank, we can always remove the redundant rows ($Ax = b$ has a solution as assumed). Besides, it only requires the existence of a feasible solution in the interior of $\mathcal{X}$ with a neighborhood that is feasible for the black-box constraints. Consequently, we have the following lemma to guarantee that the image of the affine function can cover an infinity-norm ball, which will be useful for proving the main result.

**Lemma 1** *There exists $\rho > 0$, such that $\mathcal{B}_\rho^{l,\infty}[0] \subset A\mathcal{B}_{\tilde{\rho}}^n[\tilde{x}] - b$, where*

$$\mathcal{B}_\rho^{l,\infty}[0] := \{y \in \mathbb{R}^l | \|y\|_\infty \le \rho\}, \ \ and \ \ A\mathcal{B}_{\tilde{\rho}}^n[\tilde{x}] - b := \{Ax - b | x \in \mathcal{B}_{\tilde{\rho}}^n[\tilde{x}]\}.$$

Without further notice, the proofs of all theoretical results in this paper are deferred to the appendix.

## 3 PRELIMINARIES

Before we present our solution, some preliminaries are introduced for further discussion.

### 3.1 PERFORMANCE METRIC

The sample sequences are compared to the constrained optimal solution $x^\star$ of problem (1). Similar to (Yu et al., 2017; Zhou & Ji, 2022; Ghosh et al., 2022), we are interested in three metrics,

$$\mathcal{R}_T = \sum_{t=1}^T \left(f(x^t) - f(x^\star)\right), \mathcal{V}_T = \left\|\left[\sum_{t=1}^T g(x^t)\right]^+\right\|, \ and \ \mathcal{S}_T = \left\|\sum_{t=1}^T \left(Ax^t - b\right)\right\|, \tag{4}$$

which are the cumulative regret compared to the constrained optimal solutions, the cumulative black-box constraint violations, and the cumulative violation of the affine constraints $\sum_{i=1}^N A_i x_i = b$, termed as the cumulative *shift* of $\sum_{i=1}^N A_i x_i^t$ compared to the desired $b$. The form of $\mathcal{V}_T$ is the violation of cumulative constraint value. $\mathcal{V}_T/T$ gives the violation of the average constraint value, which is common in practice when the constraint function $g_i$ represents some resource or cost that is additive over the time horizon. For example, when $g$ represents some economic

cost such as monetary expenses or energy consumption, it is usually of more interest to bound the cumulative or average constraint value during a period rather than the violation accumulated (that is, $\left\| \sum_{t=1}^{T} \left[ \sum_{i=1}^{N} g_i(x_i^t) \right]^{+} \right\|$). The same rationale also applies to the cumulative shift term $\mathcal{S}_T$. For example, in the optimal power allocation problem for wireless communication (Tse, 1997), where we assign power $p_i$ to each communication channel $i$ from fixed power budget $P$, $\sum_{t=1}^{T} \left( \sum_{i=1}^{N} p_i - P \right)$ measures the energy consumption deviation from a predefined budget, since the summation of power represents energy consumption.

### 3.2 GAUSSIAN PROCESS REGRESSION

As common in the existing Bayesian optimization methods, we use Gaussian process surrogates to learn the black-box functions. Same as in (Chowdhury & Gopalan, 2017a), we artificially introduce a set of Gaussian processes $\mathcal{GP}(0, k_{i,0}(\cdot, \cdot)), i \in [N]$ for the surrogate modeling of the unknown black-box objective function $f_i, i \in [N]$. We also adopt an i.i.d Gaussian zero-mean noise model with noise variance $\lambda > 0$, which can be chosen by the algorithm. We use the following notations,

$$k_{i,0}(x_i^{1:t}, x_i) := [k_{i,0}(x_i^1, x_i), k_{i,0}(x_i^2, x_i), \cdots, k_{i,0}(x_i^t, x_i)]^{\top},$$

$$K_{i,0}^t := (k_{i,0}(x_i^{\tau_1}, x_i^{\tau_2}))_{\tau_1 \in [t], \tau_2 \in [t]}, \text{ and } y_{i,0}^{1:t} := [y_{i,0}^1, y_{i,0}^2, \cdots, y_{i,0}^t]^{\top}.$$

We introduce the following functions of $(x_i, x_i')$,

$$\mu_{i,0}^t(x_i) = k_{i,0}(x_i^{1:t}, x_i)^{\top} \left( K_{i,0}^t + \lambda I \right)^{-1} y_{i,0}^{1:t}, \tag{5a}$$

$$k_{i,0}^t(x_i, x_i') = k_{i,0}(x_i, x_i') - k_{i,0}(x_i^{1:t}, x_i)^{\top} \left( K_{i,0}^t + \lambda I \right)^{-1} k_{i,0}(x_i^{1:t}, x_i'), \tag{5b}$$

and $\left( \sigma_{i,0}^t(x_i) \right)^2 = k_{i,0}^t(x_i, x_i)$. Similarly, we can get $\mu_{i,j}^t(\cdot), k_{i,j}^t(\cdot, \cdot), \sigma_{i,j}^t(\cdot), \forall i \in [N], \forall j \in [m]$ for the constraint function $g_{i,j}$. To characterize the complexity of the Gaussian processes and the corresponding RKHSs, we further introduce the maximum information gain for learning the objective $f_i$ as in (Srinivas et al., 2012),

$$\gamma_{i,0}^t := \max_{A \subset \mathcal{X}^i; |A| = t} \frac{1}{2} \log \left| I + \lambda^{-1} K_{i,0}^A \right|, \tag{6}$$

where $K_{i,0}^A = (k_{i,0}(x_i, x_i'))_{x_i, x_i' \in A}$. Similarly, we introduce $\gamma_{i,j}^t, \forall i \in [N], j \in [m]$ for $g_{i,j}$.

**Remark 1** *Note that the Gaussian process model here is* only *used to derive the posterior mean functions, the covariance functions, and the maximum information gain for the purpose of algorithm description and theoretical analysis. It does not change our set-up that all the black-box functions considered are deterministic functions and that the observation noise only needs to be sub-Gaussian.*

Based on the aforementioned preliminaries of Gaussian process regression, we then derive the lower confidence and upper confidence bound functions. Without further notice, all the following results are conditioned on the event in Lem. 2 happening.

**Lemma 2** *Let Assumptions 1 and 2 hold. With probability at least $1 - \delta, \forall \delta \in (0, 1)$, the following holds for all $x_i \in \mathcal{X}^i, \forall t \geq 1$, and $\forall i \in [N]$,*

$$f_i(x_i) \in [\underline{f}_i^t(x_i), \bar{f}_i^t(x_i)], \text{ and } g_{i,j}(x_i) \in [\underline{g}_{i,j}^t(x_i), \bar{g}_{i,j}^t(x_i)], \forall j \in [m], \tag{7}$$

*where for all $i \in [N], j \in [m]$,*

$$\underline{f}_i^t(x_i) := \max\{\mu_{i,0}^{t-1}(x_i) - \beta_{i,0}^t \sigma_{i,0}^{t-1}(x_i), -C_{i,0}\}, \quad \bar{f}_i^t(x_i) := \min\{\mu_{i,0}^{t-1}(x_i) + \beta_{i,0}^t \sigma_{i,0}^{t-1}(x_i), C_{i,0}\},$$

$$\underline{g}_{i,j}^t(x_i) := \max\{\mu_{i,j}^{t-1}(x_i) - \beta_{i,j}^t \sigma_{i,j}^{t-1}(x_i), -C_{i,j}\}, \quad \bar{g}_{i,j}^t(x_i) := \min\{\mu_{i,j}^{t-1}(x_i) + \beta_{i,j}^t \sigma_{i,j}^{t-1}(x_i), C_{i,j}\},$$

*with $\beta_{i,j}^t := C_{i,j} + \sigma \sqrt{2 \left( \gamma_{i,j}^{t-1} + 1 + \ln(N(m+1)/\delta) \right)}$.*

## 4 ALGORITHM AND THEORETICAL GUARANTEES

The design of our algorithm combines the celebrated ideas of GP-UCB (Srinivas et al., 2012)(lower confidence bound in our case) and dual decomposition (Boyd et al., 2007). The key idea here is

relaxing both the black-box and affine constraints, which gives the Lagrangian,

$$\mathcal{L}(x, \lambda, \mu) = \sum_{i=1}^{N} f_i(x_i) + \eta \lambda^\top \left( \sum_{i=1}^{N} g_i(x_i) \right) + \eta \mu^\top \left( \sum_{i=1}^{N} A_i x_i - b \right), \tag{9}$$

where $\eta$ is a scaling constant. Rearranging the Eq. (9) gives,

$$\mathcal{L}(x, \lambda, \mu) = \sum_{i=1}^{N} \left( f_i(x_i) + \eta \lambda^\top g_i(x_i) + \eta \mu^\top A_i x_i \right) - \eta \mu^\top b. \tag{10}$$

Then the coupled optimization problem in (1) is decomposed into local problem for each agent.

$$\min_{x \in \mathcal{X}} \mathcal{L}(x, \lambda, \mu) = \sum_{i=1}^{N} \min_{x_i \in \mathcal{X}^i} \left( f_i(x_i) + \eta \lambda^\top g_i(x_i) + \eta \mu^\top A_i x_i \right) - \eta \mu^\top b. \tag{11}$$

However, since $f_i$ and $g_i$ are both black-box functions, the local optimization problem $\min_{x_i \in \mathcal{X}^i} \left( f_i(x_i) + \eta \lambda^\top g_i(x_i) + \eta \mu^\top A_i x_i \right)$ can not be solved directly. Instead, we adopt the optimistic idea and propose to solve the local optimistic problem for agent $i$ at time step $t$,

$$\min_{x_i \in \mathcal{X}^i} \left( \underline{f}_i^t(x_i) + \eta \lambda^\top \underline{g}_i^t(x_i) + \eta \mu^\top A_i x_i \right), \tag{12}$$

where $\underline{g}_i^t(x_i) := (\underline{g}_{i,j}^t(x_i))_{j=1}^{m}$. For the dual update, we adopt the classical dual ascent method (e.g., in (Luo & Tseng, 1993)). Our primal-dual algorithm is shown in Alg. 1, where $\eta > 0$ is to be set, $0 < \epsilon \le \frac{\xi}{2}$ is a slackness parameter, and $[\cdot]^+ := \max\{\cdot, 0\}$ is interpreted element-wise.

---

**Algorithm 1** **D**istributed **M**ulti-**A**gent **B**ayesian **O**ptimization with Constraints (**DMABO**).

---

1: **for** $t \in [T]$ **do**
2:     **Local Primal update:**
$$x_i^t \in \arg \min_{x_i \in \mathcal{X}^i} \left\{ \underline{f}_i^t(x_i) + \eta \lambda_t^\top \underline{g}_i^t(x_i) + \eta \mu_t^\top A_i x_i \right\}, \forall i \in [N]. \tag{13}$$
3:     **Global Dual update:**
$$\lambda_{t+1} = [\lambda_t + \sum_{i=1}^{N} \underline{g}_i^t(x_i^t) + \epsilon e]^+, \text{ and } \mu_{t+1} = \mu_t + \sum_{i=1}^{N} A_i x_i^t - b. \tag{14}$$
4:     For each agent $i$, evaluate $f_i$ and $g_{i,j}, j \in [m]$ at $x_i^t$ with noise in a distributed way.
5:     Update $(\mu_{i,j}^t, \sigma_{i,j}^t), i \in [N], j \in \{0\} \cup [m]$ with the new data.
6: **end for**

---

Intuitively, the larger $\eta$ is, the more emphasis is given to the constraints. $\eta$ can also be interpreted as equivalent to stepsize for dual ascent. For the convenience of algorithm description and theoretical analysis, $\eta$ is set to be the same for all the constraints. Nevertheless, all the results still hold as long as $\eta$s for different constraints are of the same order ($\Theta(1/\sqrt{T})$ as will be seen in Thm. 1.).

**Remark 2 (Communication Scheme for Dual Update)** *In line 3 of the Alg. 1, the dual update is done by a central coordinator that collects $(A_i x_i^t, \underline{g}_i^t(x_i^t))$ information globally. However, this is for the generic setting in which the coupled black-box constraint takes summation over all the agents. If the black-box constraint only takes sum over a small subset of agents, then only communication over this subset of agents is needed. The same argument applies to the affine constraints. In practice, the affine constraints usually represent the consensus among the agents, and the corresponding dual variables only need to be updated in a local neighborhood.*

**Remark 3 (Dual Interpretations)** *In Alg. 1, $\lambda_t$ and $\mu_t$ are not exactly the dual variables, but the dual variables scaled by $\frac{1}{\eta}$. Indeed, $\lambda_t$ can be interpreted as virtual queue length (Zhou & Ji, 2022). The intuition of $\epsilon$ is to introduce a constant pessimistic drift to control the cumulative violation.*

### 4.1 BOUNDING CUMULATIVE REGRET/VIOLATION/SHIFT.

We now give the theoretical guarantees on the cumulative regret/violation/shift bounds in Thm. 1.

**Theorem 1** *Let the Assumptions 1–6 hold. We further assume $\lim_{T\to\infty}\sum_{i=1}^{N}\sum_{j=0}^{m}\gamma_{i,j}^{T}/\sqrt{T}=0$. We set $\eta = 1/\sqrt{T}$ [1], $\lambda_1 = \sqrt{H_1/me}, \mu_1 = 0$ and set $H_1 := 1/2\left(4C_0/(\eta\xi) + \left(4\|C\|^2+2B^2\right)/\xi\right)^2$, $H_2 := 4C_0^2/(\rho^2\eta^2)\left(1+\sqrt{m}\right)^2 + (1+\sqrt{m})^2/\rho^2\left(2\|C\|^2 + B^2\right)^2$, $C := (C_1,\cdots,C_m)$, $B := \max_{x\in\mathcal{X}}\|Ax - b\|$, $\beta_i^T := (\beta_{i,1}^T,\cdots,\beta_{i,m}^T)$, and $\gamma_i^T := (\gamma_{i,1}^T,\cdots,\gamma_{i,m}^T)$. We have,*

1. *If we set $\epsilon = \epsilon_1 := \left(\sqrt{2(H_1+H_2+\frac{2C_0}{\eta}+2\|C\|^2+B^2)}+8\sum_{i=1}^{N}\|\beta_i^T\|\sqrt{T}\|\gamma_i^T\|\right)/T$, and let $T$ be large enough such that $\epsilon = \mathcal{O}\left(\sum_{i=1}^{N}\sum_{j=0}^{m}\gamma_{i,j}^T/\sqrt{T}\right) \leq \min\left\{\xi/2, \min_{j\in[m]}C_j\right\}$. We have*

$$\mathcal{R}_T = \tilde{\mathcal{O}}\left(N\sum_{i=1}^{N}\sum_{j=0}^{m}\gamma_{i,j}^T\sqrt{T} + N^2\sqrt{T}\right), \; \mathcal{S}_T = \mathcal{O}(N\sqrt{T}) \; and \; \mathcal{V}_T=0,$$

   *where $\tilde{\mathcal{O}}(\cdot)$ hides logarithmic factor with respect to $N$ and $T$.*

2. *Alternatively, if we set $\epsilon = \epsilon_2 := \sqrt{2(H_1+H_2+\frac{2C_0}{\eta}+2\|C\|^2+B^2)}/T$, and let $T$ be large enough such that $\epsilon = \mathcal{O}\left(N/\sqrt{T}\right) \leq \min\left\{\xi/2, \min_{j\in[m]}C_j\right\}$. Then,*

$$\mathcal{R}_T = \tilde{\mathcal{O}}\left(\sum_{i=1}^{N}\gamma_{i,0}^T\sqrt{T} + N^2\sqrt{T}\right), \; \mathcal{S}_T = \mathcal{O}(N\sqrt{T}) \; and \; \mathcal{V}_T=\tilde{\mathcal{O}}\left(\sum_{i=1}^{N}\sum_{j=0}^{m}\gamma_{i,j}^T\sqrt{T}\right).$$

With the assumption $\lim_{T\to\infty}\sum_{i=1}^{N}\sum_{j=0}^{m}\gamma_{i,j}^T/\sqrt{T} = 0$, Thm. 1 shows sublinear bounds in $T$ for cumulative regret, cumulative violations, and cumulative shift for affine constraints. Thus, we have as $T \to \infty$, $\mathcal{R}_T/T \to 0$, $\mathcal{S}_T/T \to 0$, and $\mathcal{V}_T/T \to 0$. That is, our algorithm simultaneously achieves the three goals of *no-regret*, *no-violation*, and *no-shift* asymptotically. Another interesting observation is that while the bound on $\mathcal{R}_T$ has a quadratic dependency on $N$, the bound on $\mathcal{S}_T$ only has a linear dependency on $N$. Thm. 1 also shows that with smaller $\epsilon$, we can trade violation for smaller regret. As compared to (Zhou & Ji, 2022), Thm. (1) explicitly expresses the dependency on $N$ and bounds the shift $\mathcal{S}_T$. We discuss more detailed differentiations and significance of Thm. 1 in Appendix A. Specifically, when all the black-box objective and constraint functions come from RKHS with the same type of kernel functions, we observe that $\mathcal{R}_T = \tilde{\mathcal{O}}(N^2 m\gamma^T\sqrt{T} + N^2\sqrt{T})$ with $\epsilon = \epsilon_1$. If we reduce $\epsilon$ to $\epsilon_2 < \epsilon_1$, the cumulative regret bound is decreased to $\tilde{\mathcal{O}}(N\gamma^T\sqrt{T} + N^2\sqrt{T})$ while the cumulative violation is increased to $\tilde{\mathcal{O}}(Nm\gamma^T\sqrt{T})$ from 0.

**Remark 4** *In Thm. 1, we make one additional assumption that $\lim_{T\to\infty}\sum_{i=1}^{N}\sum_{j=0}^{m}\gamma_{i,j}^T/\sqrt{T} = 0$. Intuitively, it limits the complexity of the corresponding RKHS so that the maximum information gain grows slower than $\sqrt{T}$. It holds for most popular kernels, including Squared Exponential kernel and Mátern kernel (under the condition that the smoothness parameter $\nu > d/2$, where d is the input dimension. ) (Srinivas et al., 2012; Vakili et al., 2021).*

### 4.2 Conditional Strong Violation Bounds and Best-Iterate Convergence

Similar to (Zhou & Ji, 2022), $\mathcal{V}_T$ only captures the violation of the cumulative constraint value, and the Thm. 1 does not necessarily imply convergence to the static optimal solution. Hence, we further introduce the strong violation metric, $\mathcal{V}_T^+ = \sum_{t=1}^{T}\left[\sum_{i=1}^{N}g_i(x^t)\right]^+$. For general instances, it is possible that the sample sequence of the Alg. 1 oscillates and $\mathcal{V}_T^+ = \Theta(T)$ (See a simple example in the Appendix B.). This section focuses on the case with only one black-box constraint and no affine constraint, which is common in many resource allocation problems, to show conditions under which we can further bound the strong violation and guarantee the best-iterate convergence. We fix $\epsilon = \epsilon_1$. The results can easily be extended to the case with multiple black-box constraints and $\epsilon = \epsilon_2$.

**Condition 1** *There exists $\alpha > 0$ and $\bar{r} > 0$, such that $\forall\pi \in \Pi(\mathcal{X})$, $\forall 0 < r \leq \bar{r}$ satisfying $\mathbb{E}_\pi[f(x)] \leq f(x^\star) + r$ and $\mathbb{E}_\pi[g(x)] \leq r$, we have $\mathbb{E}_\pi[|g(x)|] \leq \alpha r$.*

---

[1]In Thm. 1, the choice of $\eta$ assumes the knowledge of $T$. We can apply the doubling trick (Besson & Kaufmann, 2018) to get the bounds without knowing $T$ beforehand (similar for $\epsilon$).

Condition 1 captures the case where $g(x^\star) = 0$ is active, and the constraint contradicts the objective (e.g., in optimal power allocation for wireless communication). To achieve $r$-optimal solution, the constraint is expected to be close to tight and not oscillating too much (analogous to dissipativity (Müller, 2021), where oscillation causes loss/dissipation to the objective function $f$).

**Condition 2** *There exists $\zeta > 0$, such that $\forall x \in \mathcal{X}$ satisfying $g(x) > 0$, we have $f(x) > f(x^\star) + \zeta$.*

Condition 2 captures the case where the constraint $g(x^\star) < 0$ is inactive and infeasible points have strictly worse objectives than the optimal feasible solution. If $f$ and $g$ are sampled from independent and symmetric Gaussian processes, it *holds with probability* $1/2$ from a Bayesian point of view. The bounds on the strong violation and the best-iterate convergence are then given in Thm. 2. It highlights that under not uncommon conditions, our algorithm also performs well in terms of managing the strong violations and finding the static constrained optimal solution.

**Theorem 2** *Let the same assumptions as in Thm. 1 hold. We further assume $m = 1, \epsilon = \epsilon_1$ and no affine constraint exists. We have,*

1. *Under Condition 1, $\mathcal{V}_T^+ = \tilde{\mathcal{O}} \left( N \sum_{i=1}^N \sum_{j=0}^m \gamma_{i,j}^T \sqrt{T} + N^2 \sqrt{T} \right).$*

2. *Under Condition 2, $\mathcal{V}_T^+ = \tilde{\mathcal{O}} \left( N^2 \sum_{i=1}^N \sum_{j=0}^m \gamma_{i,j}^T \sqrt{T} + N^3 \sqrt{T} \right).$ Furthermore, there exists $T_0 > 0$, such that $\forall T \geq T_0$, there exists $\tilde{x}^T \in \{x^1, \cdots, x^T\}$, which satisfies,*

$$\sum_{i=1}^N \left( f_i(\tilde{x}_i^T) - f_i(x_i^\star) \right) = \tilde{\mathcal{O}} \left( \frac{N^2 \sum_{i=1}^N \sum_{j=0}^m \gamma_{i,j}^T + N^3}{\sqrt{T}} \right), \quad and \quad \sum_{i=1}^N g_i(\tilde{x}_i^T) \leq 0.$$

## 5 EXPERIMENTS

Two sets of experiments are conducted to demonstrate the performance of the DMABO algorithm. In the first set, we use the objective and constraint functions sampled from Gaussian processes without affine constraints. In the second set, we consider a more realistic optimal power allocation problem for wireless communication (Tse, 1997). We compare our method to the distributed simultaneous version of the CEI (Gelbart et al., 2014; Gardner et al., 2014) algorithm, where in each step, each agent maximizes the constrained expected improvement conditioned on the decisions of other agents fixed as in the last step. We also compare our method to the heuristic multi-agent Bayesian optimization method (Krishnamoorthy & Paulson, 2023), where a global coordinator assigns a penalty to the local acquisition step. We refer the readers to our appendix and the attached code for more details (choice of (hyper-)parameters, computational time and performance metrics, etc.).

### 5.1 SAMPLED INSTANCES FROM GAUSSIAN PROCESSES

We first consider the scenario without affine constraint. Such a setting arises widely in a variety of real-world applications. For example, in demand response for a smart grid (Chen et al., 2018), one may want to maximize the total utilities for multiple consumers while controlling their total energy consumption below some threshold. We set $N = 3, m = 2$, and $\mathcal{X}^i = [-1, 1] \subset \mathbb{R}, \forall i \in [3]$.

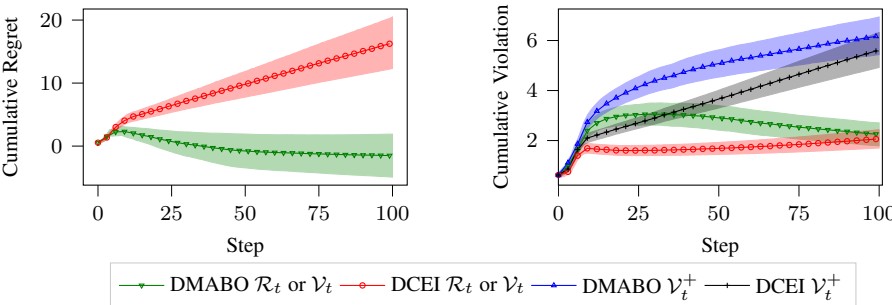

Figure 1: Cumulative regret $\mathcal{R}_t$ and violation $\mathcal{V}_t$ averaged over 100 random instances. The shaded area represents $\pm 0.2$ standard deviation for regret and $\pm 0.1$ standard deviation for violation.

The black-box functions are sampled from Gaussian processes with the squared exponential kernel. Fig. 1 shows the cumulative regret and violation result. It can be seen that our DMABO algorithm clearly achieves a sublinear growth rate for most of the cases, and for many cases, our DMABO algorithm even achieves better performance than the static optimal solution (that is, regret $\leq 0$) while controlling the cumulative violation well. Note that the decrease in cumulative violation is due to the 'compensation' effect. In contrast, the oblivious distributed extension of the CEI algorithm (DCEI) suffers from linear regret growth with growing violations. For DMABO, the strong violation $\mathcal{V}^+$ clearly grows slower and slower, while DCEI suffers from linear growth.

## 5.2 OPTIMAL POWER ALLOCATION FOR WIRELESS COMMUNICATION

In this part, we consider the classic optimal power allocation problem (Tse, 1997) for wireless communication. Mathematically, we aim to solve the following optimization problem,

$$\min_{p_i \in [p_i^{\min}, p_i^{\max}]} -\sum_{i=1}^{N} U_i(p_i), \quad \text{subject to:} \quad \sum_{i=1}^{N} p_i = P, \tag{15}$$

where $U_i : \mathbb{R} \to \mathbb{R}$ is the utility function (that measures, e.g., quality of service, or communication rate) of the agent $i$. Here, the dual variable $\mu$ corresponding to the constraint $\sum_{i=1}^{N} p_i = P$ can be interpreted as the power price. We compare our DMABO algorithm to the heuristic algorithm (Krishnamoorthy & Paulson, 2023). Specifically, in each step, we penalize the EI acquisition function (Jones et al., 1998) by a quadratic penalty function of the difference with respect to the co-ordinated power computed with an ADMM type method (Krishnamoorthy & Paulson, 2023). Fig. 2 shows the average utility and cumulative power deviation from the power budget. Our DMABO algorithm achieves $8.4\%$ higher average utility with $78.1\%$ less cumulative power deviation as compared to the penalty heuristics with a penalty $5$. In this example, further increasing the penalty improves the power deviation only very slightly.

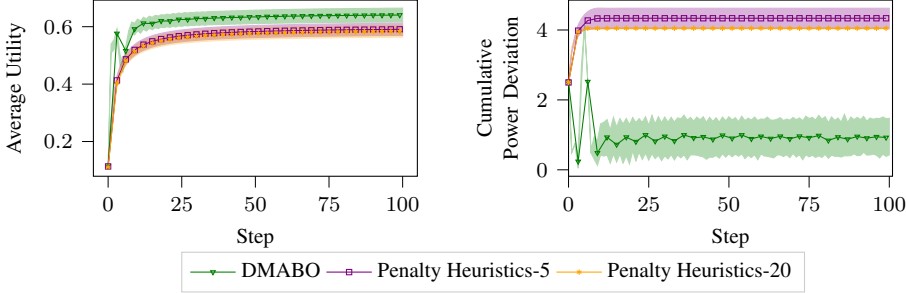

Figure 2: The average utility and the cumulative power deviation $|\sum_{\tau=1}^{t}(\sum_{i=1}^{N} p_i^{\tau} - P)|$, which measures the deviation of total power compared to the budget $P$, of the two algorithms. 'Penalty Heuristics-$Q$' represents the penalty method with penalty term $Q$.

## 6 CONCLUSION AND FUTURE WORK

In this paper, we have studied the problem of distributed multi-agent Bayesian optimization, with both coupled black-box constraints and known affine constraints. We propose a primal-dual distributed algorithm with similar regret/violation bounds as those in the single-agent case for the black-box objective and constraint functions. Furthermore, the algorithm guarantees an $\mathcal{O}(N\sqrt{T})$ bound on the cumulative violation for the known affine constraints, ensuring that the average of the historical samples satisfies the affine constraints up to the error $\mathcal{O}(N/\sqrt{T})$. We also characterize mild conditions under which the strong violation can be bounded, and best-iterate convergence is guaranteed. The method is then applied to both sampled instances from Gaussian processes and real-world experimental examples; the results show that the method simultaneously provides close-to-optimal performance and maintains minor violations on average, corroborating our theoretical analysis. As for future work, one direction is reducing the dependency of regret on the number of agents ($N^2$ in this paper).

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

# A  APPENDIX: PROOF OF THM. 1

This appendix gives detailed proof of the Thm. 1. Without further notice, all the results are under the same assumptions as the Thm. 1. Intuitively, the regret/violation has two sources. The first source comes from the learning cost to reduce the uncertainties of the black-box functions, measured by the posterior standard deviations. The second source is due to the nature of the primal-dual algorithm.

Our proof is mainly inspired by (Zhou & Ji, 2022). However, as compared to (Zhou & Ji, 2022), we have additional affine constraints to deal with. How to incorporate and bound the violations for the affine constraints makes our analysis more challenging and different from (Zhou & Ji, 2022). As will be seen, we construct a similar potential function in the dual variables corresponding to both the black-box constraints and the affine constraints. However, we can not use the same technique to bound the potential function as in (Zhou & Ji, 2022). Instead, we will separately bound the two parts in the potential function, which then leads to one bound on the sum of the two parts.

As a reminder, throughout the appendix, we will use the following notations,

$$f(x) := \sum_{i=1}^{N} f_i(x_i), \ \underline{f}^t(x) := \sum_{i=1}^{N} \underline{f}_i^t(x_i), \tag{16a}$$

$$g(x) := \sum_{i=1}^{N} g_i(x_i), \ \underline{g}^t(x) := \sum_{i=1}^{N} \underline{g}_i^t(x_i). \tag{16b}$$

Before we prove the main theorem, we prove several useful lemmas. We first give the proof of Lem. 1.

**Lemma 1** *There exists $\rho > 0$, such that $\mathcal{B}_\rho^{l,\infty}[0] \subset A\mathcal{B}_{\tilde{\rho}}^n[\tilde{x}] - b$, where*

$$\mathcal{B}_\rho^{l,\infty}[0] := \{y \in \mathbb{R}^l \,|\, \|y\|_\infty \leq \rho\}, \quad and \quad A\mathcal{B}_{\tilde{\rho}}^n[\tilde{x}] - b := \{Ax - b | x \in \mathcal{B}_{\tilde{\rho}}^n[\tilde{x}]\}.$$

**Proof:**  Since $A$ is full row rank by Assump. 6, there exists $l$ columns of $A$ that forms an invertible submatrix. Without loss of generality, we assume the first $l$ columns of $A$ forms an invertible matrix. We use $A_l$ to denote the submatrix formed by the first $l$ columns of $A$. We set $\rho = \tilde{\rho}/\|A_l^{-1}\|_{\infty,2}$, where $\|A_l^{-1}\|_{\infty,2} := \max_{\|y\|_\infty \leq 1} \|A_l^{-1}y\| > 0$. For any $y \in \mathcal{B}_\rho^{l,\infty}[0]$, we have,

$$A\left(\tilde{x} + [A_l^{-1}y; 0_{n-l}]\right) - b = A\tilde{x} - b + A[A_l^{-1}y; 0_l] = y,$$

where $[A_l^{-1}y; 0_{n-l}]$ represents the vertical concatenation of the vector $A_l^{-1}y$ and the vertical vector $0_{n-l}$ consisting of $n - l$ 0s. We also have,

$$\|[A_l^{-1}y; 0_{n-l}]\| = \|A_l^{-1}y\| \leq \|A_l^{-1}\|_{\infty,2}\|y\|_\infty \leq \|A_l^{-1}\|_{\infty,2}\rho = \tilde{\rho}.$$

Therefore, $y = A\left(\tilde{x} + [A_l^{-1}y; 0_{n-l}]\right) - b \in A\mathcal{B}_{\tilde{\rho}}^n[\tilde{x}] - b$.  □

We then give the proof of Lem. 2.

**Lemma 2** *Let Assumptions 1 and 2 hold. With probability at least $1 - \delta, \forall \delta \in (0, 1)$, the following holds for all $x_i \in \mathcal{X}^i, \forall t \geq 1$, and $\forall i \in [N]$,*

$$f_i(x_i) \in [\underline{f}_i^t(x_i), \bar{f}_i^t(x_i)], \ \text{ and } \ g_{i,j}(x_i) \in [\underline{g}_{i,j}^t(x_i), \bar{g}_{i,j}^t(x_i)], \ \forall j \in [m], \tag{7}$$

*where for all $i \in [N], j \in [m]$,*

$$\underline{f}_i^t(x_i) := \max\{\mu_{i,0}^{t-1}(x_i) - \beta_{i,0}^t\sigma_{i,0}^{t-1}(x_i), -C_{i,0}\}, \ \bar{f}_i^t(x_i) := \min\{\mu_{i,0}^{t-1}(x_i) + \beta_{i,0}^t\sigma_{i,0}^{t-1}(x_i), C_{i,0}\},$$

$$\underline{g}_{i,j}^t(x_i) := \max\{\mu_{i,j}^{t-1}(x_i) - \beta_{i,j}^t\sigma_{i,j}^{t-1}(x_i), -C_{i,j}\}, \ \bar{g}_{i,j}^t(x_i) := \min\{\mu_{i,j}^{t-1}(x_i) + \beta_{i,j}^t\sigma_{i,j}^{t-1}(x_i), C_{i,j}\},$$

*with $\beta_{i,j}^t := C_{i,j} + \sigma\sqrt{2\left(\gamma_{i,j}^{t-1} + 1 + \ln(N(m+1)/\delta)\right)}$.*

**Proof:**  By Corollary 2.6, (Xu et al., 2023a), with probability at least $1 - \delta, \forall \delta \in (0, 1)$, for all $x_i \in \mathcal{X}^i$ and $t \geq 1$,

$$\mu_{i,j}^{t-1}(x_i) - \beta_{i,j}^t\sigma_{i,j}^{t-1}(x_i) \leq g_{i,j}(x_i) \leq \mu_{i,j}^{t-1}(x_i) + \beta_{i,j}^t\sigma_{i,j}^{t-1}(x_i).$$

Furthermore, $|g_{i,j}(x_i)| = |\langle g_{i,j}, k_{i,j}(x_i, \cdot)\rangle| \le \|g_{i,j}\| \|k_{i,j}(x_i, \cdot)\| = \|g_{i,j}\| k_{i,j}(x_i, x_i) \le C_{i,j}, \forall i \in [N], j \in [m]$. Therefore, $g_{i,j}(x_i) \in [\underline{g}_{i,j}^t(x_i), \bar{g}_{i,j}^t(x_i)]$. Similarly, $f_i(x_i) \in [\underline{f}_i^t(x_i), \bar{f}_i^t(x_i)], \forall i \in [N]$. $\qquad\square$

We then restate a useful lemma that bounds the cumulative standard deviations along the sample trajectory.

**Lemma 3 (Lemma 4, (Chowdhury & Gopalan, 2017b))** *Given a sequence of points* $x^1, x^2, \cdots, x^T$ *from* $\mathcal{X}$*, we have,*

$$\sum_{t=1}^T \sigma_{i,j}^{t-1}\left(x_i^t\right) \le \sqrt{4(T+2)\gamma_{i,j}^T}, \forall i \in [N], j \in \{0\} \cup [m]. \tag{17}$$

To characterize how optimality can be traded for more feasibility, we introduce the perturbed problem,

$$\min_{\pi \in \Pi(\mathcal{X})} \quad \mathbb{E}_\pi\left[f(x)\right], \tag{18a}$$

$$\text{subject to:} \quad \mathbb{E}_\pi\left[g(x)\right] + \epsilon e \le 0, \tag{18b}$$

$$\mathbb{E}_\pi\left[Ax\right] = b, \tag{18c}$$

where the feasible set is relaxed to the set of all distributions over the set $\mathcal{X}$. Such a relaxation results in a linear programming problem in distribution, which is easier for sensitivity analysis. We use $\pi_\epsilon^*$ to denote the optimal solution to the above problem. We then have the following lemma.

**Lemma 4**

$$\sum_{t=1}^T \mathbb{E}_{\pi_\epsilon^*}\left[f(x)\right] - \sum_{t=1}^T \mathbb{E}_{\pi^\star}\left[f(x)\right] \le \frac{2C_0 T \epsilon}{\xi}, \tag{19}$$

*where* $\pi^\star$ *is the optimal distribution for Problem (18) with* $\epsilon = 0$.

**Proof:** Let $\pi_\epsilon = (1 - \frac{\epsilon}{\xi})\pi^\star + \frac{\epsilon}{\xi}\bar{\pi}$, where $\mathbb{E}_{\bar{\pi}}\left[g(x)\right] \le -\xi e$ (Recall the Assump. 5). We also have $\mathbb{E}_{\pi^\star}\left[g(x)\right] \le 0$. Then

$$\mathbb{E}_{\pi_\epsilon}\left[g(x)\right] = \left(1 - \frac{\epsilon}{\xi}\right)\mathbb{E}_{\pi^\star}\left[g(x)\right] + \frac{\epsilon}{\xi}\mathbb{E}_{\bar{\pi}}\left[g(x)\right]$$

$$\le -\epsilon e.$$

Furthermore, by linearity,

$$\mathbb{E}_{\pi_\epsilon}\left[Ax\right] = \left(1 - \frac{\epsilon}{\xi}\right)\mathbb{E}_{\pi^\star}\left[Ax\right] + \frac{\epsilon}{\xi}\mathbb{E}_{\bar{\pi}}\left[Ax\right]$$

$$= b.$$

Hence, $\pi_\epsilon$ is a feasible solution to the slightly perturbed problem. So, we have,

$$\sum_{t=1}^T \mathbb{E}_{\pi_\epsilon^\star}\left[f(x)\right] - \sum_{t=1}^T \mathbb{E}_{\pi^\star}\left[f(x)\right]$$

$$\le \sum_{t=1}^T \mathbb{E}_{\pi_\epsilon}\left[f(x)\right] - \sum_{t=1}^T \mathbb{E}_{\pi^\star}\left[f(x)\right]$$

$$= \frac{\epsilon}{\xi}\sum_{t=1}^T \left(\mathbb{E}_{\bar{\pi}}\left[f(x)\right] - \mathbb{E}_{\pi^\star}\left[f(x)\right]\right)$$

$$\le \frac{2C_0 T \epsilon}{\xi},$$

where the first inequality follows by the optimality of $\pi_\epsilon^*$, the equality follows by the definition of $\pi_\epsilon$ and the last inequality follows by Assumption 2, which implies that

$$|f(x)| = |\sum_{i=1}^{N}\langle f_i, k_{i,0}(x_i,\cdot)\rangle| \leq \sum_{i=1}^{N}|\langle f_i, k_{i,0}(x_i,\cdot)\rangle| \leq \sum_{i=1}^{N}\|f_i\|\|k_{i,0}(x_i,\cdot)\| \leq \sum_{i=1}^{N}\|f_i\| \leq C_0.$$

$\square$

It will be seen that $\epsilon$ plays a key role in trading some regret for strict time-average feasibility.

To connect the change of the dual variables and the primal values, we introduce a potential function in the dual space,

$$V(\lambda_t, \mu_t) = \frac{1}{2}\|\lambda_t\|^2 + \frac{1}{2}\|\mu_t\|^2. \tag{20}$$

We consider,

$$\Delta_t := V(\lambda_{t+1}, \mu_{t+1}) - V(\lambda_t, \mu_t) \tag{21a}$$

$$= \frac{1}{2}\left(\|[\lambda_t + \underline{g}^t(x^t) + \epsilon e]^+\|^2 - \|\lambda_t\|^2\right) + \frac{1}{2}\|Ax^t - b\|^2 + \mu_t^\top(Ax^t - b) \tag{21b}$$

$$\leq \frac{1}{2}\left(\|\lambda_t + \underline{g}^t(x^t) + \epsilon e\|^2 - \|\lambda_t\|^2\right) + \frac{1}{2}\|Ax^t - b\|^2 + \mu_t^\top(Ax^t - b) \tag{21c}$$

$$= \lambda_t^\top(\underline{g}^t(x^t) + \epsilon e) + \mu_t^\top(Ax^t - b) + \frac{1}{2}\|\underline{g}^t(x^t) + \epsilon e\|^2 + \frac{1}{2}\|Ax^t - b\|^2, \tag{21d}$$

where the inequality follows by case discussion on the sign of $\lambda_t + \underline{g}^t(x^t) + \epsilon e$. The inequality in (21) will be very useful in the following proof by connecting primal and dual variables.

### A.0.1 Bound Cumulative Regret

We have the following lemma to bound $\underline{f}^t(x^t) - \mathbb{E}_{\pi_\epsilon^*}\left[\underline{f}^t(x)\right]$, which approximates the single-step instantaneous regret $f(x^t) - f(x^\star)$.

**Lemma 5**

$$\underline{f}^t(x^t) - \mathbb{E}_{\pi_\epsilon^*}\left[\underline{f}^t(x)\right] \leq 2\eta\|C\|^2 + \eta B^2 - \eta\Delta_t,$$

where $\|C\|^2 = \sum_{j=1}^{m}C_j^2$, $\underline{f}^t(x) = \sum_{i=1}^{N}\underline{f}_i^t(x_i)$, and $\epsilon$ is set to be small enough such that $\epsilon \leq C_j, \forall j \in [m]$.

**Proof:**

$$\Delta_t = V(\lambda_{t+1}, \mu_{t+1}) - V(\lambda_t, \mu_t) \tag{22a}$$

$$\leq \lambda_t^\top(\underline{g}^t(x^t) + \epsilon e) + \mu_t^\top(Ax^t - b) + \frac{1}{2}\|\underline{g}^t(x^t) + \epsilon e\|^2 + \frac{1}{2}\|Ax^t - b\|^2 \tag{22b}$$

$$\leq \lambda_t^\top \underline{g}^t(x^t) + \frac{1}{\eta}\underline{f}^t(x^t) + \mu_t^\top(Ax^t - b) + \epsilon\lambda_t^\top e - \frac{1}{\eta}\underline{f}^t(x^t) + \frac{1}{2}\sum_{j=1}^{m}(C_j + \epsilon)^2 + \frac{1}{2}B^2 \tag{22c}$$

$$\leq \lambda_t^\top \mathbb{E}_{\pi_\epsilon^*}\left[\underline{g}^t(x)\right] + \frac{1}{\eta}\mathbb{E}_{\pi_\epsilon^*}\left[\underline{f}^t(x)\right] + \epsilon\lambda_t^\top e - \frac{1}{\eta}\underline{f}^t(x^t) + 2\sum_{j=1}^{m}C_j^2 + B^2 \tag{22d}$$

$$\leq \lambda_t^\top\left(\mathbb{E}_{\pi_\epsilon^*}[g(x)] + \epsilon e\right) + \frac{1}{\eta}\mathbb{E}_{\pi_\epsilon^*}\left[\underline{f}^t(x)\right] - \frac{1}{\eta}\underline{f}^t(x^t) + 2\sum_{j=1}^{m}C_j^2 + B^2 \tag{22e}$$

$$\leq \frac{1}{\eta}\left(\mathbb{E}_{\pi_\epsilon^*}\left[\underline{f}^t(x)\right] - \underline{f}^t(x^t)\right) + 2\|C\|^2 + B^2, \tag{22f}$$

where the first inequality follows by the inequality (21), the second inequality follows by adding and subtracting $\frac{1}{\eta}\underline{f}^t(x^t)$ and the projection operation to $[-C_i, C_i]$ as shown in the Eq. (8), the third inequality follows by the optimality of $x^t$ for the primal update problem (13) and the assumption that $\epsilon \leq C_j$, the fourth inequality follows by $\underline{g}^t(x) \leq g(x)$ and the feasibility of $\pi_\epsilon^\star$ for the problem (18),

and the last inequality follows by the feasibility of $\pi_\epsilon^*$ for the problem (18). Rearrangement of the above inequality gives the desired result. □

We are then ready to upper bound the cumulative regret.

**Lemma 6 (Cumulative Regret Bound)**

$$\mathcal{R}_T \leq 2 \sum_{i=1}^{N} \beta_{i,0}^T \sqrt{4(T+2)\gamma_{i,0}^T} + 2\eta T \|C\|^2 + \eta T B^2 + \eta V(\lambda_1, \mu_1) + \frac{2C_0 T \epsilon}{\xi}.$$

**Proof:** We do the relaxation and splitting,

$$\mathcal{R}_T \leq \sum_{t=1}^{T} \left( f(x^t) - \mathbb{E}_{\pi^\star}[f(x)] \right)$$

$$= \sum_{t=1}^{T} \left( f(x^t) - \underline{f}^t(x^t) \right) + \sum_{t=1}^{T} \left( \underline{f}^t(x^t) - \mathbb{E}_{\pi_\epsilon^\star}\left[\underline{f}^t(x^t)\right] \right)$$

$$+ \sum_{t=1}^{T} \mathbb{E}_{\pi_\epsilon^\star}\left[\underline{f}^t(x) - f(x)\right] + \sum_{t=1}^{T} \left( \mathbb{E}_{\pi_\epsilon^\star}[f(x)] - \mathbb{E}_{\pi^\star}[f(x)] \right),$$

where the inequality follows by that relaxed optimal value $\mathbb{E}_{\pi^\star}[f(x)]$ is smaller or equal to the original optimal value, and the equality splits the original term into four terms. For the first term,

$$\sum_{t=1}^{T} \left( f(x^t) - \underline{f}^t(x^t) \right) = \sum_{t=1}^{T} \sum_{i=1}^{N} \left( f_i(x_i^t) - \underline{f}_i^t(x_i^t) \right) \leq \sum_{i=1}^{N} \sum_{t=1}^{T} 2\beta_{i,0}^t \sigma_{i,0}^t(x_i^t)$$

$$\leq 2 \sum_{i=1}^{N} \beta_{i,0}^T \sum_{t=1}^{T} \sigma_{i,0}^t(x_i^t) \leq 2 \sum_{i=1}^{N} \beta_{i,0}^T \sqrt{4(T+2)\gamma_{i,0}^T},$$

where the first inequality follows by Lem. 2, the second inequality follows by the monotonicity of $\beta_{i,0}^t$, and the last inequality follows by Lem. 3. For the second term, by Lem. 5, we have,

$$\sum_{t=1}^{T} \left( \underline{f}^t(x^t) - \mathbb{E}_{\pi_\epsilon^\star}\left[\underline{f}^t(x)\right] \right)$$

$$\leq \sum_{t=1}^{T} (2\eta\|C\|^2 + \eta B^2 - \eta\Delta_t)$$

$$= 2\eta T \|C\|^2 + \eta T B^2 + \eta V(\lambda_1, \mu_1) - \eta V(\lambda_{T+1}, \mu_{T+1})$$

$$\leq 2\eta T \|C\|^2 + \eta T B^2 + \eta V(\lambda_1, \mu_1).$$

The third term is non-positive due to Lem. 2. Combining the three bounds and the Lem. 4 gives the desired result. □

### A.0.2 BOUND CUMULATIVE VIOLATION

The dual update indicates that violations are reflected in the dual variable. So we first upper bound the dual variable. The idea is to show that whenever the dual variable is very large, it will be decreased. We now separate the dual potential function $V(\lambda, \mu)$ into two parts, $V_1(\lambda) = \frac{1}{2}\|\lambda\|^2$ and $V_2(\mu) = \frac{1}{2}\|\mu\|^2$. It can thus be seen that $V(\lambda_t, \mu_t) = V_1(\lambda_t) + V_2(\mu_t)$.

**Lemma 7** *If $V(\lambda_t, \mu_t) \geq H_1 + H_2$, we have $V(\lambda_{t+1}, \mu_{t+1}) \leq V(\lambda_t, \mu_t)$.*

**Proof:** We prove the lemma by discussing different cases.

**Case 1**: $V_1(\lambda_t) \geq H_1 = \frac{1}{2}\left(\frac{4C_0}{\eta\xi} + \frac{4\|C\|^2 + 2B^2}{\xi}\right)^2$.

By the primal updating rule,

$$\underline{f}^t(x^t) + \eta\lambda_t^\top \underline{g}^t(x^t) + \eta\mu_t^\top(Ax^t - b)$$

$$\leq \mathbb{E}_{\bar{\pi}} \left[ \underline{f}^t(x) \right] + \eta \lambda_t^\top \mathbb{E}_{\bar{\pi}} \left[ \underline{g}^t(x) \right]$$

$$\leq C_0 + \eta \lambda_t^\top \mathbb{E}_{\bar{\pi}} \left[ g(x) \right]$$

$$\leq C_0 + \eta(-\xi) \lambda_t^\top e,$$

where the first inequality follows by the optimality of $x^t$ for the primal update problem and the feasibility of $\bar{\pi}$ for the affine constraint, the second inequality follows by that both $\eta$ and $\lambda_t$ are non-negative, and the third inequality follows by Lem. 2 and Assumption 5. On the other hand,

$$\underline{f}_t(x_t) \geq -C_0,$$

by the Lem. 2. Therefore,

$$C_0 + \eta(-\xi) \lambda_t^\top e \geq -C_0 + \eta \lambda_t^\top \underline{g}^t(x^t) + \eta \mu_t^\top (Ax^t - b),$$

which implies

$$\lambda_t^\top \underline{g}^t(x^t) + \mu_t^\top (Ax^t - b) \leq \frac{2C_0}{\eta} - \xi \lambda_t^\top e.$$

So we can get

$$V(\lambda_{t+1}, \mu_{t+1}) - V(\lambda_t, \mu_t)$$

$$\leq \lambda_t^\top (\underline{g}^t(x^t) + \epsilon e) + \mu_t^\top (Ax^t - b) + \frac{1}{2} \| \underline{g}_t(x^t) + \epsilon e \|^2 + \frac{1}{2} \| Ax_t - b \|^2$$

$$\leq \frac{2C_0}{\eta} - \frac{\xi}{2} \lambda_t^\top e + \frac{1}{2} \| \underline{g}^t(x^t) + \epsilon e \|^2 + \frac{1}{2} \| Ax^t - b \|^2$$

$$\leq \frac{2C_0}{\eta} - \frac{\xi}{2} \| \lambda_t \| + 2 \| C \|^2 + B^2 \leq 0,$$

where the first inequality follows by the inequality (21), the second inequality follows by that $\epsilon \leq \frac{\xi}{2}$, the third inequality follows by that $\lambda_t \geq 0$ and the Lem. 2, and the last inequality follows by $V_1(\lambda_t) \geq \frac{1}{2} \left( \frac{4C_0}{\eta\xi} + \frac{4\|C\|^2 + 2B^2}{\xi} \right)^2$.

**Case 2:** $V_1(\lambda_t) < H_1 = \frac{1}{2} \left( \frac{4C_0}{\eta\xi} + \frac{4\|C\|^2 + 2B^2}{\xi} \right)^2$. By Lem. 1, there exists $x(\mu_t) \in \mathcal{B}_\rho^n[\tilde{x}]$, such that $Ax(\mu_t) - b = -\rho \text{sign}(\mu_t)$, where $\forall k \in [l]$,

$$(\text{sign}(\mu_t))_k = \begin{cases} 1, & \text{if } (\mu_t)_k \geq 0, \\ -1, & \text{otherwise.} \end{cases}$$

By the primal updating rule, we have

$$\underline{f}^t(x^t) + \eta \lambda_t^\top \underline{g}^t(x^t) + \eta \mu_t^\top (Ax^t - b)$$

$$\leq \underline{f}^t(x(\mu_t)) + \eta \lambda_t^\top \underline{g}^t(x(\mu_t)) - \eta \rho \mu_t^\top \text{sign}(\mu_t)$$

$$\leq C_0 + \eta \lambda_t^\top g(x(\mu_t)) - \eta \rho \| \mu_t \|_1$$

$$\leq C_0 - \eta \rho \| \mu_t \|.$$

Meanwhile, we also have,

$$\underline{f}^t(x^t) \geq -C_0. \tag{23}$$

Therefore,

$$-C_0 + \eta \lambda_t^\top \underline{g}^t(x^t) + \eta \mu_t^\top (Ax^t - b) \leq C_0 - \eta \rho \| \mu_t \|$$

Rearrangement gives,

$$\lambda_t^\top \underline{g}^t(x^t) + \mu_t^\top (Ax^t - b) \leq \frac{2C_0}{\eta} - \rho \| \mu_t \|.$$

Therefore, we can derive,

$$V(\lambda_{t+1}, \mu_{t+1}) - V(\lambda_t, \mu_t) \tag{24}$$

$$\leq \lambda_t^\top (\underline{g}^t(x^t) + \epsilon e) + \mu_t^\top (Ax^t - b) + \frac{1}{2} \| \underline{g}^t(x^t) + \epsilon e \|^2 + \frac{1}{2} \| Ax^t - b \|^2 \tag{25}$$

$$\leq \frac{2C_0}{\eta} + \frac{\xi\sqrt{m}}{2}\|\lambda_t\| - \rho\|\mu_t\| + \frac{1}{2}\|\underline{g}^t(x^t) + \epsilon e\|^2 + \frac{1}{2}\|Ax^t - b\|^2 \tag{26}$$

$$\leq \frac{2C_0}{\eta} + \frac{\xi\sqrt{m}}{2}\|\lambda_t\| - \rho\|\mu_t\| + 2\|C\|^2 + B^2 \tag{27}$$

$$\leq \frac{2C_0}{\eta} + \frac{\xi\sqrt{m}}{2}\left(\frac{4C_0}{\eta\xi} + \frac{4\|C\|^2 + 2B^2}{\xi}\right) - \rho\|\mu_t\| + 2\|C\|^2 + B^2 \tag{28}$$

$$= \frac{2C_0}{\eta}\left(1 + \sqrt{m}\right) + \left(1 + \sqrt{m}\right)\left(2\|C\|^2 + B^2\right) - \rho\|\mu_t\|. \tag{29}$$

Since $V_1(\lambda_t) < H_1, V_1(\lambda_t) + V_2(\mu_t) \geq H_1 + H_2$, we have $V_2(\mu_t) \geq H_2$. Hence, the last term in the inequality (29) can be checked to be non-positive.

Combining the two cases concludes the proof. $\qquad\square$

Consequently, we have,

**Lemma 8** *Let* $\lambda_1 \leq \sqrt{\frac{H_1}{N}}, \mu_1 = 0$, *we have for any* $t$, $V(\lambda_t, \mu_t) \leq H_1 + H_2 + \frac{2C_0}{\eta} + 2\|C\|^2 + B^2$.

**Proof:** We use induction. $(\lambda_1, \mu_1)$ satisfies the conclusion.

We now discuss conditioned on the value $(\lambda_t, \mu_t)$. If for step $t$, $V(\lambda_t, \mu_t) \leq H_1 + H_2 + \frac{2C_0}{\eta} + 2\|C\|^2 + B^2$ holds. Then there are two possible cases,

**Case 1**: $V(\lambda_t, \mu_t) \leq H_1 + H_2$, then

$$V(\lambda_{t+1}, \mu_{t+1}) \leq V(\lambda_t, \mu_t) + \frac{1}{\eta}\left(\mathbb{E}_{\pi_\epsilon^*}\left[\underline{f}^t(x)\right] - \underline{f}^t(x^t)\right) + 2\|C\|^2 + B^2$$

$$\leq H_1 + H_2 + \frac{2C_0}{\eta} + 2\|C\|^2 + B^2$$

by Lem. 5.

**Case 2**: $V(\lambda_t, \mu_t) > H_1 + H_2$, then $V(\lambda_{t+1}, \mu_{t+1}) \leq V(\lambda_t, \mu_t) \leq H_1 + H_2 + \frac{2C_0}{\eta} + 2\|C\|^2 + B^2$ by Lem. 7.

By induction, the conclusion holds for any $t$. $\qquad\square$

We now can upper bound the cumulative violation.

**Lemma 9 (Cumulative Violation Bound)**

$$\mathcal{V}_T \leq \left\|\left[\lambda_{T+1} + 2\sum_{i=1}^{N}\beta_i^T\sqrt{4(T+2)\gamma_i^T} - T\epsilon e\right]^+\right\|,$$

*where* $\beta_i^T = (\beta_{i,1}^T, \cdots, \beta_{i,m}^T)$, $\gamma_i^T = (\gamma_{i,1}^T, \cdots, \gamma_{i,m}^T), i \in [N]$ *and multiplication is interpreted elementwise.*

**Proof:** By the dual updating rule, we have $\lambda_{t+1} \geq \lambda_t + \underline{g}^t(x^t) + \epsilon e$. By summing up from $t = 1$ to $T$, we get,

$$\lambda_{T+1} \geq \lambda_1 + \sum_{t=1}^{T}\underline{g}^t(x^t) + T\epsilon e.$$

Rearranging the above inequality gives,

$$\sum_{t=1}^{T}\underline{g}^t(x^t) \leq \lambda_{T+1} - \lambda_1 - T\epsilon e. \tag{30}$$

We thus have,

$$\sum_{t=1}^{T}g(x^t) = \sum_{t=1}^{T}\underline{g}^t(x^t) + \sum_{t=1}^{T}(g(x^t) - \underline{g}^t(x^t))$$

$$= \sum_{t=1}^{T} \underline{g}^t(x^t) + \sum_{t=1}^{T} \sum_{i=1}^{N} (g_i(x_i^t) - \underline{g}_i^t(x_i^t))$$

$$\leq \lambda_{T+1} - \lambda_1 - T\epsilon e + 2 \sum_{i=1}^{N} \beta_i^T \sqrt{4(T+2)\gamma_i^T},$$

where the inequality follows by combining the inequality (30), the monotonicity of $\beta_i^T$ and Lem. 3.. Therefore,

$$\mathcal{V}_T = \left\| \left[ \sum_{t=1}^{T} g(x^t) \right]^+ \right\| \leq \left\| \left[ \lambda_{T+1} + 2 \sum_{i=1}^{N} \beta_i^T \sqrt{4(T+2)\gamma_i^T} - T\epsilon e \right]^+ \right\|.$$

$\square$

### A.0.3 BOUND CUMULATIVE SHIFT

**Lemma 10** $\left\| \sum_{t=1}^{T} (Ax^t - b) \right\| \leq \|\mu_{T+1}\| + \|\mu_1\|.$

**Proof:**

$$\left\| \sum_{t=1}^{T} (Ax^t - b) \right\|$$

$$= \left\| \sum_{t=1}^{T} (\mu_{t+1} - \mu_t) \right\|$$

$$= \|\mu_{T+1} - \mu_1\|$$

$$\leq \|\mu_{T+1}\| + \|\mu_1\|.$$

$\square$

### A.1 MAIN PROOF OF THM. 1

Note that $1/\eta = \mathcal{O}(\sqrt{T})$ and $C_j = \sum_{i=1}^{N} C_{i,j} = \mathcal{O}(N), \forall j \in \{0\} \cup [m]$. Firstly, combining Lem. 8 and Lem. 10 gives,

$$\mathcal{S}_T = \mathcal{O}\left( N\sqrt{T} \right).$$

We then discuss different selections of $\epsilon$.

1. If we set

$$\epsilon = \frac{\sqrt{2(H_1 + H_2 + \frac{2C_0}{\eta} + 2\|C\|^2 + B^2)} + 8\sum_{i=1}^{N} \|\beta_i^T\| \sqrt{T\|\gamma_i^T\|}}{T},$$

and let $T$ be large enough such that

$$\epsilon = \tilde{\mathcal{O}}\left( \sum_{i=1}^{N} \sum_{j=0}^{m} \gamma_{i,j}^T / \sqrt{T} \right) \leq \min\left\{ \xi/2, \min_{j \in [m]} C_j \right\}.$$

We have,

$$\lambda_{T+1} + 2 \sum_{i=1}^{N} \beta_i^T \sqrt{4(T+2)\gamma_i^T} - T\epsilon e \tag{31a}$$

$$\leq \left( \|\lambda_{T+1}\| + 8 \sum_{i=1}^{N} \|\beta_i^T\| \sqrt{T\|\gamma_i^T\|} - T\epsilon \right) e \tag{31b}$$

$$\leq \left( \sqrt{2(H_1 + H_2 + \frac{2C_0}{\eta} + 2\|C\|^2 + B^2)} + 8 \sum_{i=1}^{N} \|\beta_i^T\| \sqrt{T\|\gamma_i^T\|} - T\epsilon \right) e \tag{31c}$$

$$=0, \tag{31d}$$

where the first inequality follows by simple algebraic manipulation, and the second inequality follows by Lem. 8. Combining the above inequality and Lem. 9 gives

$$\mathcal{V}_T = 0.$$

Plugging the values of $\eta, \lambda_1$ and $\epsilon$ into the Lem. 6 gives,

$$\mathcal{R}_T = \tilde{\mathcal{O}} \left( N \sum_{i=1}^{N} \sum_{j=0}^{m} \gamma_{i,j}^T \sqrt{T} + N^2 \sqrt{T} \right).$$

2. If we set

$$\epsilon = \frac{\sqrt{2(H_1 + H_2 + \frac{2C_0}{\eta} + 2\|C\|^2 + B^2)}}{T},$$

and let $T$ be large enough such that

$$\epsilon = \mathcal{O}\left(N/\sqrt{T}\right) \le \min\left\{\xi/2, \min_{j\in[m]} C_j\right\}.$$

We have,

$$\lambda_{T+1} + 2 \sum_{i=1}^{N} \beta_i^T \sqrt{4(T+2)\gamma_i^T} - T\epsilon e \tag{32a}$$

$$\le \left( \sqrt{2(H_1 + H_2 + \frac{2C_0}{\eta} + 2\|C\|^2 + B^2)} - T\epsilon \right) e + 2 \sum_{i=1}^{N} \beta_i^T \sqrt{4(T+2)\gamma_i^T} \tag{32b}$$

$$\le 2 \sum_{i=1}^{N} \beta_i^T \sqrt{4(T+2)\gamma_i^T}, \tag{32c}$$

where the first inequality follows by Lem. 8. Combining the above inequality and Lem. 9 gives

$$\mathcal{V}_T = \tilde{\mathcal{O}} \left( \sum_{i=1}^{N} \sum_{j=0}^{m} \gamma_{i,j}^T \sqrt{T} \right).$$

Plugging the values of $\eta, \lambda_1$ and $\epsilon$ into the Lem. 6 gives,

$$\mathcal{R}_T = \tilde{\mathcal{O}} \left( \sum_{i=1}^{N} \gamma_{i,0}^T \sqrt{T} + N^2 \sqrt{T} \right).$$

## B APPENDIX: AN ILLUSTRATIVE EXAMPLE WHERE THE SEQUENCES GENERATED BY PRIMAL-DUAL ALGORITHM OSCILLATE

We consider one single agent and one single black-box constraint with one-dimensional input. We set $\mathcal{X} = \{-1, 0, 1\}$ with the corresponding objective and constraint function shown as in Tab. 1.

| $x$ | $f(x)$ | $g(x)$ |
|-----|--------|--------|
| $-1$ | $1$ | $-1$ |
| $0$ | $0.5$ | $0$ |
| $1$ | $-1$ | $2$ |

Table 1: Illustrative counter-example where the primal solution oscillates, and convergence is never achieved.

Suppose there is no observation noise, and we have already successfully identified all the black-box function values after some observation. Then, $\mathcal{L}(x, \lambda) = f(x) + \lambda g(x)$. We observe,

$$\min\{\mathcal{L}(-1, \lambda), \mathcal{L}(1, \lambda)\}$$
$$\le \frac{2}{3}\mathcal{L}(-1, \lambda) + \frac{1}{3}\mathcal{L}(1, \lambda) = \frac{2}{3}(1 - \eta\lambda) + \frac{1}{3}(-1 + 2\eta\lambda) = \frac{1}{3} < 0.5 = \mathcal{L}(0, \lambda).$$

So, the primal-dual algorithm will never sample the point $x = 0$, which is the optimal solution for the constrained optimization problem. Instead, when $\lambda \leq \frac{2}{3\eta}$, $\arg\min_{x \in \mathcal{X}} \mathcal{L}(x, \lambda) = \{1\}$, and $\lambda$ will be increased. Otherwise, $\arg\min_{x \in \mathcal{X}} \mathcal{L}(x, \lambda) = \{-1\}$, $\lambda$ will be decreased. So the sample sequence oscillates in $\{-1, 1\}$, with about $1/3$ proportion as 1 and the other $-1$ to stabilize $\lambda$ around $\frac{2}{3\eta}$. This results in $\Theta(T)$ growth for the cumulative strong violation $\sum_{t=1}^{T} [g(x^t)]^+$.

## C    APPENDIX: PROOF OF THM. 2.

We will use the short-hand notation $[\cdot]^- = -\min\{0, \cdot\}$ and $\mathbb{E}_T$ to represent the expectation over the empirical uniform distribution over the sample set $\{x^1, \cdots, x^T\}$. We will also use the notations,
$$\mathcal{T}^+ = \{t \in [T] | g(x^t) > 0\}, \text{ and } \mathcal{T}^- = \{t \in [T] | g(x^t) \leq 0\}.$$
It can be seen that $\mathcal{T}^+ \cup \mathcal{T}^- = [T]$.

### C.1    PROOF UNDER CONDITION 1

Firstly, by Thm. 1,

$$\frac{\mathcal{R}_T}{T} = \sum_{t=1}^{T} \frac{1}{T} f(x^t) - f(x^\star) \tag{33a}$$

$$= \mathbb{E}_T [f(x)] - f(x^\star) \tag{33b}$$

$$= \tilde{\mathcal{O}} \left( \frac{N \sum_{i=1}^{N} \sum_{j=0}^{m} \gamma_{i,j}^T + N^2}{\sqrt{T}} \right), \tag{33c}$$

$$\tag{33d}$$

and

$$\sum_{t=1}^{T} g(x^t) = T \mathbb{E}_T [g(x)] \leq 0. \tag{34}$$

Let $T$ be large enough such that,

$$\mathbb{E}_T [f(x)] - f(x^\star) = \tilde{\mathcal{O}} \left( \frac{N \sum_{i=1}^{N} \sum_{j=0}^{m} \gamma_{i,j}^T + N^2}{\sqrt{T}} \right) \leq \bar{r}. \tag{35}$$

Therefore, by Condition 1, we have,

$$\mathbb{E}_T [|g(x)|] \leq \alpha \left( \mathbb{E}_T [f(x)] - f(x^\star) \right) = \alpha \frac{\mathcal{R}_T}{T}. \tag{36}$$

Hence,

$$\sum_{t=1}^{T} [g(x^t)]^+ + \sum_{t=1}^{T} [g(x^t)]^- \leq \alpha \mathcal{R}_T. \tag{37}$$

Furthermore, by Eq. (34),

$$\sum_{t=1}^{T} g(x^t) = \sum_{t=1}^{T} [g(x^t)]^+ - \sum_{t=1}^{T} [g(x^t)]^- \leq 0. \tag{38}$$

Adding the Inequality (37) and the Inequality (38) gives,

$$\mathcal{V}_T^+ = \sum_{t=1}^{T} [g(x^t)]^+ \leq \frac{\alpha}{2} \mathcal{R}_T = \tilde{\mathcal{O}} \left( N \sum_{i=1}^{N} \sum_{j=0}^{m} \gamma_{i,j}^T \sqrt{T} + N^2 \sqrt{T} \right). \tag{39}$$

### C.2    PROOF UNDER CONDITION 2

We have,

$$\mathcal{R}_T = \sum_{t=1}^{T} \left( f(x^t) - f(x^\star) \right) \tag{40a}$$

$$= \sum_{t \in \mathcal{T}^+} \left( f(x^t) - f(x^\star) \right) + \sum_{t \in \mathcal{T}^-} \left( f(x^t) - f(x^\star) \right) \tag{40b}$$

$$\geq \sum_{t \in \mathcal{T}^+} \left( f(x^t) - f(x^\star) \right) \tag{40c}$$

$$\geq \zeta |\mathcal{T}^+|. \tag{40d}$$

Therefore, $|\mathcal{T}^+| \leq \mathcal{R}_T / \zeta$. Thus,

$$\mathcal{V}_T^+ = \sum_{t=1}^T [g(x^t)]^+ \tag{41a}$$

$$= \sum_{t \in \mathcal{T}^+} [g(x^t)]^+ \tag{41b}$$

$$\leq \sum_{t \in \mathcal{T}^+} C_1 \tag{41c}$$

$$= |\mathcal{T}^+| C_1 \tag{41d}$$

$$\leq \frac{C_1}{\zeta} \mathcal{R}_T. \tag{41e}$$

With $\epsilon = \epsilon_1$, we have

$$\mathcal{R}_T = \tilde{\mathcal{O}} \left( N \sum_{i=1}^N \sum_{j=0}^m \gamma_{i,j}^T \sqrt{T} + N^2 \sqrt{T} \right), \quad \text{and} \quad \mathcal{V}_T^+ = \tilde{\mathcal{O}} \left( N^2 \sum_{i=1}^N \sum_{j=0}^m \gamma_{i,j}^T \sqrt{T} + N^3 \sqrt{T} \right),$$

by Thm 1 and the inequality (41). Therefore,

$$\sum_{t=1}^T \left( \left( f(x^t) - f(x^\star) \right) + [g(x^t)]^+ \right) \tag{42a}$$

$$= \mathcal{R}_T + \mathcal{V}_T^+ = \tilde{\mathcal{O}} \left( N^2 \sum_{i=1}^N \sum_{j=0}^m \gamma_{i,j}^T \sqrt{T} + N^3 \sqrt{T} \right). \tag{42b}$$

Hence,

$$\frac{1}{T} \sum_{t=1}^T \left( \left( f(x^t) - f(x^\star) \right) + [g(x^t)]^+ \right) = \tilde{\mathcal{O}} \left( \frac{N^2 \sum_{i=1}^N \sum_{j=0}^m \gamma_{i,j}^T + N^3}{\sqrt{T}} \right). \tag{43}$$

Choose $T_0$ large enough such that $\forall T \geq T_0$, we have,

$$\frac{1}{T} \sum_{t=1}^T \left( \left( f(x^t) - f(x^\star) \right) + [g(x^t)]^+ \right) = \tilde{\mathcal{O}} \left( \frac{N^2 \sum_{i=1}^N \sum_{j=0}^m \gamma_{i,j}^T + N^3}{\sqrt{T}} \right) < \zeta. \tag{44}$$

Hence, there exists $\tilde{x}^T \in \{x^1, \cdots, x^T\}$, such that,

$$\left( f(\tilde{x}^T) - f(x^\star) \right) + [g(\tilde{x}^T)]^+ \tag{45a}$$

$$\leq \frac{1}{T} \sum_{t=1}^T \left( \left( f(x^t) - f(x^\star) \right) + [g(x^t)]^+ \right) \tag{45b}$$

$$= \tilde{\mathcal{O}} \left( \frac{N^2 \sum_{i=1}^N \sum_{j=0}^m \gamma_{i,j}^T + N^3}{\sqrt{T}} \right) \tag{45c}$$

$$< \zeta. \tag{45d}$$

It can be observed that $g(\tilde{x}^T) \leq 0$, otherwise, by Condition 2, $\left( f(\tilde{x}^T) - f(x^\star) \right) + [g(\tilde{x}^T)]^+ > \zeta$, which contradicts the inequality (45d). Furthermore,

$$f(\tilde{x}^T) - f(x^\star) \leq \left( \left( f(\tilde{x}^T) - f(x^\star) \right) + [g(\tilde{x}^T)]^+ \right) \leq \tilde{\mathcal{O}} \left( \frac{N^2 \sum_{i=1}^N \sum_{j=0}^m \gamma_{i,j}^T + N^3}{\sqrt{T}} \right). \tag{46}$$

# D    APPENDIX: MORE DETAILS ON THE EXPERIMENT

The experiments are implemented in python, based on the package GPy (GPy, since 2012). **Choice of (Hyper-)parameters.** The performance of DMABO (same as general GP-UCB/LCB algorithm) is mainly impacted by the choice of confidence bound coefficient $\beta_{i,j}^t$. For sampled instances from the Gaussian process, we set $\beta_{i,j}^t$ according to the theoretical analysis. In real-world practice, $\beta_{i,j}^t$ can usually be set as a constant. Indeed, when the kernel choices and the kernel hyperparameters fit the black-box functions well, setting $\beta_{i,j}^t = 3$ typically works well. In our power allocation example, manually setting $\beta_{i,j}^t = 3.0$ works well empirically. We also set $\lambda = 0.02^2$ for the Gaussian process modeling. We use the common squared exponential kernel functions.

**Computational Time.** In our experiments, the local decision variables all have low-dimensional inputs ($n_i \leq 3$). So we use pure grid search to solve the local primal update problem, which is relatively cheap as compared to the evaluations of the typical ground-truth functions in practice due to the known expressions of the lower confidence bound functions.

**Performance Metrics.** To measure the performance of different algorithms, we use the regret $\mathcal{R}_t$ shown in Eq. (4). To measure the violations, we use the violation of the cumulative black-box constraint value $\mathcal{V}_t = \left\| \left[ \sum_{\tau=1}^t \sum_{i=1}^N g_{i,j}(x_i^\tau) \right]^+ \right\|$ and the cumulative violations for affine constraints $\mathcal{S}_t = \left\| \sum_{\tau=1}^t \left( \sum_{i=1}^N A_i x_i^\tau - b \right) \right\|$.

