# OpenReview forum: "Multi-Agent Bayesian Optimization with Coupled Black-box and Affine Constraints"
_ICLR.cc/2024/Conference — Submitted to ICLR 2024_

### Official Review · Reviewer_yaEw · 2023-10-31

**Soundness:** 3 good
**Presentation:** 3 good
**Contribution:** 2 fair
**Rating:** 6
**Confidence:** 4

**Summary:**

This paper studies an extension of a recent paper by [Zhou and Ji (NeurIPS 2022)](https://proceedings.neurips.cc/paper_files/paper/2022/hash/00295cede6e1600d344b5cd6d9fd4640-Abstract-Conference.html) on constrained Gaussian process optimization (equivalently, constrained black-box bayesian optimization, constrained kernelized bandit optimization). In particular, the extension assumes that the objective $f$ is given by $f(x) = \sum_{I=1}^N f_i(x_i)$ where $x=(x_1,\ldots, x_N)^\top$ and the constraint function $g$ is given by $g(x) = \sum_{I=1}^N g_i(x_i)$ where $x=(x_1,\ldots, x_N)^\top$. The setting here is referred to as *multi-agent* in the paper in that individual agent $i\in [N]$ makes its decision $x_i$. Moreover, there exists an affine constraint $\sum_{i=1}^N A_i x_i = b$ that couples the individual decisions $x_1,\ldots, x_N$. Here, the basic idea is to dualize the affine constraint to decouple the decisions $x_1,\ldots, x_N$. Then one may apply the drift-based primal-dual method of [Zhou and Ji (NeurIPS 2022)](https://proceedings.neurips.cc/paper_files/paper/2022/hash/00295cede6e1600d344b5cd6d9fd4640-Abstract-Conference.html) developed for the single-agent setting to optimize the individual decisions.

**Strengths:**

* The paper proposes an extension of the existing single-agent constrained black-box optimization framework to a multi-agent setting with a joint affine constraint.
* Numerical results demonstrate the soundness of the proposed framework on some applications.

**Weaknesses:**

* Although the distributed multi-agent framework proposed in this paper has meaningful applications in practice, the technical contribution of the paper seems marginal. The main component of the algorithm developed in this paper relies on the method by [Zhou and Ji (NeurIPS 2022)](https://proceedings.neurips.cc/paper_files/paper/2022/hash/00295cede6e1600d344b5cd6d9fd4640-Abstract-Conference.html), and the primal-dual technique to deal with the joint affine constraint as well as the aggregated constraint function is standard. The most important part to derive the performance guarantees is basically done by the work of [Zhou and Ji (NeurIPS 2022)](https://proceedings.neurips.cc/paper_files/paper/2022/hash/00295cede6e1600d344b5cd6d9fd4640-Abstract-Conference.html) for the single-agent setting.
* It is a restriction that the functions of individual agents $(f_1,g_1),\ldots, (f_N, g_N)$ come from independent Gaussian processes.

**Questions:**

Is it possible to consider more complex interactions between agents? It might be a too strong assumption that the functions of individual agents $(f_1,g_1),\ldots, (f_N, g_N)$ come from independent Gaussian processes. Is it possible to study the setting where $f=(f_1,\ldots, f_N)$ and $g=(g_1,\ldots,g_N)$ follow a joint Gaussian process?

---

> ### Author Response · Authors · 2023-11-17
> **Response 3-1**
>
> > Although the distributed multi-agent framework proposed in this paper has meaningful applications in practice, the technical contribution of the paper seems marginal. The main component of the algorithm developed in this paper relies on the method by Zhou and Ji (NeurIPS 2022), and the primal-dual technique to deal with the joint affine constraint as well as the aggregated constraint function is standard. The most important part to derive the performance guarantees is basically done by the work of Zhou and Ji (NeurIPS 2022) for the single-agent setting.
>
> Thanks for the comment. This concern is similar to that of the reviewer vwM2. So we duplicate the response here.
> Indeed, our algorithm and analysis are inspired by   (Zhou \& Ji, 2022) and (Srinivas et al., 2012). However, our paper significantly differentiates from the existing works in the following three aspects.
>    * **Formulation and design** We consider a multi-agent scenario with coupled black-box and affine constraints, which widely appear in all kinds of engineering areas. This is significantly harder than the single-agent case since the black-box function evaluation and the acquisition step needs to be distributed, while the coupled constraints need coordination. Primal methods, such as EIC (Gardner et al., 2014) and CONFIG (Xu et al., 2023), are not readily applicable to the multi-agent scenario. The key of our method is combining the dual-decomposition method and the GP-UCB method, which leads to a distributed primal-dual algorithm, which, as far as we know, is the first algorithm that can tackle the constrained multi-agent Bayesian optimization problem with theoretical regret/violation guarantees.
>    * **Analysis**  The additional components of affine constraints bring new analysis challenges as compared to (Zhou \& Ji, 2022). Specifically, the regret bounds in Thm. 1 (Zhou \& Ji, 2022) is highly sensitive to the Slater condition parameter $\delta$ (indeed, $\mathcal{O}(\frac{1}{\delta})$). One may rewrite the affine constraints as two linear inequality constraints equivalently.  However, for affine constraints, such Slater condition does not hold anymore. Oblivious application of the analysis pipeline in (Zhou \& Ji, 2022) fails to give a finite bound since $\frac{1}{\delta}\to\infty$ when $\delta\to0$ for affine constraints, thus necessitating the separate analysis on these two types of constraints. To remedy this issue, we introduce a critical new technique (See Lem. 1 and the discussion on case 2 in the proof of Lem. 7), which is a non-obvious extension of the analysis in (Zhou \& Ji, 2022).
>
>    * **Conditional bounds on strong violation** Fig. 1 (c) and Fig. 2 (c) in (Zhou \& Ji, 2022) shows an empirical sublinear growth of the strong constraint violation (that is, $\sum_{t=1}^T[g(x^t)]^+$). However, (Zhou \& Ji, 2022) did not give any theoretical insights into why this sublinear growth happens even though, in many applications, it is of more importance to manage this stronger metric of violation when the constraint can not be compensated across time (e.g., thermal discomfort in building control). Indeed, we show that in general, primal-dual method can not guarantee sublinear growth of strong violation (see a counterexample in the Appendix B). Hence, there seems to be a contradiction between theory and practice. Meanwhile, existing work that can provide theoretical bound on the strong violation, such as CONFIG (Xu et al., 2023), are not applicable to the distributed multi-agent scenario, which seems to suggest that distributed algorithm and sublinear strong violation bound can not be obtained simultaneously. To fill this gap and provide theoretical insights into the empirical observations, we provide two mild and physically meaningful conditions under which the primal-dual method can simultaneously provide the sublinear strong violation bound and the distributed algorithm. Specifically, the first condition deals with the case that $f$ and $g$ are contradictory metrics (which is common in practice) and $g(x^\star)=0$, that is, the constraint is active on the optimal solution. Our condition 1 is intuitively analogous to the concept of dissipativity in system theory. A simple example that satisfies condition 1 is when $g$ is invertible and the composed function $f\circ g^{-1}$ is convex. The second condition deals with the case that $g(x^\star)<0$, that is, the constraint is inactive on the optimal solution. Intuitively, condition 2 means infeasible solution is strictly worse than the feasible optimal solution. Interestingly, condition 2 holds with probability $\frac{1}{2}$ if the objective and the constraint are sampled from two independent Gaussian processes.

---

> > ### Comment · Reviewer_yaEw · 2023-11-22
> >
> > Thank you for your response. Your clarifications about conditional bounds on strong violation make sense, which does seem to make a difference compared to the setting of Zhou and Ji (NeurIPS 2022). However, although dealing with the additional affine equality constraint was not considered by Zhou and Ji (NeurIPS 2022), the technique and analysis are common in constrained optimization. That said, I would raise the score from 5 to 6.

---

> ### Author Response · Authors · 2023-11-17
> **Response 3-2**
>
> > It is a restriction that the functions of individual agents $(f_1, g_1), \cdots, (f_N, g_N)$ come from independent Gaussian processes. Is it possible to consider more complex interactions between agents? It might be a too strong assumption that the functions of individual agents $(f_1, g_1), \cdots, (f_N, g_N)$ come from independent Gaussian processes. Is it possible to study the setting where $f=(f_1, \cdots, f_N)$ and $g=(g_1, \cdots, g_N)$ follow a joint Gaussian process?
>
> Thanks for the great suggestion. Our current setting is more like a frequentist setting, where the unknown objective/constraint is only assumed to be smooth in some sense. Independence of the functions is not assumed. To incorporate the correlated functions setting algorithmically, we need to combine the multi-output Gaussian processes with the primal-dual algorithm. Indeed, one direction to explore is combining our method with the federated Bayesian optimization method.

---

> ### Author Response · Authors · 2023-11-21
> **Final question**
>
> Dear reviewer,
>
> Since the rebuttal will end soon, we are wondering if you have any final questions. Could you please give us any feedback, even if you did not change your mind on the rating? Your feedback will greatly help us plan our research activity (waiting for final decision or withdrawal for resubmission purpose).
>
> Many thanks,
>
> All the authors.

---

> ### Author Response · Authors · 2023-11-22
> **Thanks for the update**
>
> Dear reviewer,
>
> We are very happy to know that you find our clarifications make sense. Thank you very much for the feedback and raising the score.
>
> Best regards,
>
> All the authors

---

### Official Review · Reviewer_vwM2 · 2023-11-01

**Soundness:** 2 fair
**Presentation:** 3 good
**Contribution:** 2 fair
**Rating:** 5
**Confidence:** 5

**Summary:**

The paper studies the problem of online distributed multi-agent Bayesian optimization with both coupled black-box constraints and known affine equation constraints. To solve this problem, the author/s propose a primal-dual distributed algorithm, which is proven to achieve similar regret and constraint violation bounds as those in the single-agent case. The paper also examines certain conditions under which a stronger metric of cumulative violation is bounded and provide best-iterate convergence without affine constraint. The examples also show the effective of the proposed algorithm.

**Strengths:**

1. The proposed algorithm is the first distributed multi-agent BO algorithm that has theoretical regret and constraint violation bounds.
2. The author/s also examine certain conditions under which the proposed algorithm could upper bound the cumulative constraint violation in sub-linear.
3. There are two examples showing the algorithm's effectiveness.

**Weaknesses:**

Although the proposed algorithm is the first online distributed multi-agent BO method with theoretical regret and constraint violation bounds guarantee, both the algorithm and the analysis are direct extension of the previous algorithm in single-agent case like the Zhou & Ji, 2022 and Srinivas et al., 2012. The author/s claim that the added affine equation constraint would lead to more challenging analysis, but to the best of my knowledge after checking the proof of Theorem 1, I could not see much challenge besides analyzing the potential function of the two terms separately instead of doing it together. And this separate analysis does not look as challenging as it sounds.

=========== After author/s feedback ===============
Thanks the author/s to spend lots of time addressing my novelty concern especially the technical contribution about the analysis for the added affine equation constraint. I agree with other reviewers on the additional complexity that the affine equation constraint brings and would like to increase my score from 3 to 5.

**Questions:**

N/A

---

> ### Author Response · Authors · 2023-11-17
> **Response 2-1**
>
> > Although the proposed algorithm is the first online distributed multi-agent BO method with theoretical regret and constraint violation bounds guarantee, both the algorithm and the analysis are direct extension of the previous algorithm in single-agent case like the Zhou & Ji, 2022 and Srinivas et al., 2012. The author/s claim that the added affine equation constraint would lead to more challenging analysis, but to the best of my knowledge after checking the proof of Theorem 1, I could not see much challenge besides analyzing the potential function of the two terms separately instead of doing it together. And this separate analysis does not look as challenging as it sounds.
>
> Thanks for the comment. Indeed, our algorithm and analysis are inspired by   (Zhou \& Ji, 2022) and (Srinivas et al., 2012). However, our paper significantly differentiates from the existing works in the following three aspects.
>    * **Formulation and design** We consider a multi-agent scenario with coupled black-box and affine constraints, which widely appear in all kinds of engineering areas. This is significantly harder than the single-agent case since the black-box function evaluation and the acquisition step needs to be distributed, while the coupled constraints need coordination. Primal methods, such as EIC (Gardner et al., 2014) and CONFIG (Xu et al., 2023), are not readily applicable to the multi-agent scenario. The key of our method is combining the dual-decomposition method and the GP-UCB method, which leads to a distributed primal-dual algorithm, which, as far as we know, is the first algorithm that can tackle the constrained multi-agent Bayesian optimization problem with theoretical regret/violation guarantees.
>    * **Analysis**  The additional components of affine constraints bring new analysis challenges as compared to (Zhou \& Ji, 2022). Specifically, the regret bounds in Thm. 1 (Zhou \& Ji, 2022) is highly sensitive to the Slater condition parameter $\delta$ (indeed, $\mathcal{O}(\frac{1}{\delta})$). One may rewrite the affine constraints as two linear inequality constraints equivalently.  However, for affine constraints, such Slater condition does not hold anymore. Oblivious application of the analysis pipeline in (Zhou \& Ji, 2022) fails to give a finite bound since $\frac{1}{\delta}\to\infty$ when $\delta\to0$ for affine constraints, thus necessitating the separate analysis on these two types of constraints. To remedy this issue, we introduce a critical new technique (See Lem. 1 and the discussion on case 2 in the proof of Lem. 7), which is a non-obvious extension of the analysis in (Zhou \& Ji, 2022).
>
>    * **Conditional bounds on strong violation** Fig. 1 (c) and Fig. 2 (c) in (Zhou \& Ji, 2022) shows an empirical sublinear growth of the strong constraint violation (that is, $\sum_{t=1}^T[g(x^t)]^+$). However, (Zhou \& Ji, 2022) did not give any theoretical insights into why this sublinear growth happens even though, in many applications, it is of more importance to manage this stronger metric of violation when the constraint can not be compensated across time (e.g., thermal discomfort in building control). Indeed, we show that in general, primal-dual method can not guarantee sublinear growth of strong violation (see a counterexample in the Appendix B). Hence, there seems to be a contradiction between theory and practice. Meanwhile, existing work that can provide theoretical bound on the strong violation, such as CONFIG (Xu et al., 2023), are not applicable to the distributed multi-agent scenario, which seems to suggest that distributed algorithm and sublinear strong violation bound can not be obtained simultaneously. To fill this gap and provide theoretical insights into the empirical observations, we provide two mild and physically meaningful conditions under which the primal-dual method can simultaneously provide the sublinear strong violation bound and the distributed algorithm. Specifically, the first condition deals with the case that $f$ and $g$ are contradictory metrics (which is common in practice) and $g(x^\star)=0$, that is, the constraint is active on the optimal solution. Our condition 1 is intuitively analogous to the concept of dissipativity in system theory. A simple example that satisfies condition 1 is when $g$ is invertible and the composed function $f\circ g^{-1}$ is convex. The second condition deals with the case that $g(x^\star)<0$, that is, the constraint is inactive on the optimal solution. Intuitively, condition 2 means infeasible solution is strictly worse than the feasible optimal solution. Interestingly, condition 2 holds with probability $\frac{1}{2}$ if the objective and the constraint are sampled from two independent Gaussian processes.

---

> ### Author Response · Authors · 2023-11-21
> **Final question**
>
> Dear reviewer,
>
> Since the rebuttal will end soon, we are wondering if you have any final questions.
> Could you please give us any feedback, even if you did not change your mind on the rating?
> Your feedback will greatly help us plan our research activity (waiting for final decision or withdrawal for resubmission purpose).
>
> Many thanks,
>
> All the authors.

---

> ### Author Response · Authors · 2023-11-22
> **Final reminder**
>
> Dear reviewer,
>
> Since the rebuttal is ending very soon, could you please give us any feedback, even if you did not change your mind on the rating? Your feedback will greatly help us plan our research activity (waiting for final decision or withdrawal for resubmission purpose).
>
> If you have further questions or comments, we are also very happy to provide more clarifications to address them.
>
> Many thanks,
>
> All the authors.

---

### Official Review · Reviewer_t8mC · 2023-11-01

**Soundness:** 3 good
**Presentation:** 3 good
**Contribution:** 3 good
**Rating:** 6
**Confidence:** 4

**Summary:**

This paper proposes the first algorithm for distributed multi-agent Bayesian optimization with both black-box and known affine constraints. The proposed method is based on the primal-dual decomposition, which naturally gives rise to separated local primal update of the input queries and global dual update of the dual variables. The paper analyzes both the regret and the cumulative violations of both constraints, and applies the method to an experiment with know affine constraints.

**Strengths:**

- The paper identifies a new problem, proposes a natural algorithm based on the primal-dual formulation, and performs rigorous analysis of the algorithm in terms of both the regrets and constraint violations.
- The paper gives real-world motivating examples when introducing different components of the problem setting, such as problems where black-box constraints and know affine constraints can occur (for example, the top paragraph of page 2).

**Weaknesses:**

- There is a related line of work on federated Bayesian optimization ([1,2] below) and federated kernelized bandits [3,4] which should be discussed (perhaps in the Introduction). In fact, the problem setting of multi-agent BO shares many similarities with federated BO/kernelized bandits, including both the first point (the agents do not share the local observations) and second point (the function evaluations are done locally) discussed in the bottom paragraph of page 1. These previous works may not diminish the novelty of the current paper though, because as far as I know, no prior federated BO/kernelized bandit papers have considered constraints.
[1] Federated Bayesian Optimization via Thompson Sampling, 2020,
[2] Differentially Private Federated Bayesian Optimization with Distributed Exploration, 2021,
[3] Communication Efficient Distributed Learning for Kernelized Contextual Bandits, 2022,
[4] Kernel-Based Federated Learning with Personalization, 2022.

- The paper did give some real-world scenarios with black-box constraints and know affine constraints, however, are there use cases with both types of constraints? The experiments section also only has one experiment with only known affine constraints, so I think it will be helpful to apply the method to an experiment with both types of constraints.
- I think the example given at the top of page 5 can be further clarified.
- Theorem 1: I think some discussions regarding the differences between the two cases would be helpful. Regarding the dependency on $\sum^N_{i=1}\sum^m_{j=1}\gamma^T_{i,j}$, if we substitute in the rate of $\gamma$ for either the SE kernel and Matern kernel, I guess this term would depend linearly on $N$ and $m$? If this is the case, then for case 1 of Theorem 1, the first term in the regret bound would have a dependency of $N^2$ and $m$. Also, it is mentioned in the paragraph below Theorem 1 that "we can trade violation for smaller regret", I think some elaborations would be helpful.
- Remark 4: if I understand correctly, the assumption $\lim_{T\rightarrow \infty}\sum^N_{i=1}\sum^m_{j=1}\gamma^T_{i,j} / \sqrt{T} = 0$ holds for the Matern kernel only when the function is smooth?
- Figure 1: I agree that the strong violation of the proposed DMABO grows slower and slower yet the baseline of DCEI suffers from linear growth, but in this figure, the strong violations of DMABO is always larger than DCEI. It seems the relative magnitude is only reversed when the number of steps is large enough.

**Questions:**

I have listed some questions under the "Weaknesses" above.

---

> ### Author Response · Authors · 2023-11-17
> **Response 1-1**
>
> > There is a related line of work on federated Bayesian optimization ([1,2] below) and federated kernelized bandits [3,4] which should be discussed (perhaps in the Introduction). In fact, the problem setting of multi-agent BO shares many similarities with federated BO/kernelized bandits, including both the first point (the agents do not share the local observations) and second point (the function evaluations are done locally) discussed in the bottom paragraph of page 1. These previous works may not diminish the novelty of the current paper though, because as far as I know, no prior federated BO/kernelized bandit papers have considered constraints.
> [1] Federated Bayesian Optimization via Thompson Sampling, 2020,
> [2] Differentially Private Federated Bayesian Optimization with Distributed Exploration, 2021,
> [3] Communication Efficient Distributed Learning for Kernelized Contextual Bandits, 2022,
> [4] Kernel-Based Federated Learning with Personalization, 2022.
>
> Many thanks for pointing out this line of works to us. They are indeed very relevant to this paper and we have added some discussion on them highlighted in color blue in the introduction section.
> > The paper did give some real-world scenarios with black-box constraints and know affine constraints, however, are there use cases with both types of constraints? The experiments section also only has one experiment with only known affine constraints, so I think it will be helpful to apply the method to an experiment with both types of constraints.
>
> Thanks for the great suggestions. Yes, there are certainly use cases with both types of constraints. For example, in demand response of smart grid, the operator aims to maximize the sum of net profit from each agent, which can be modeled as black-box functions of price, subject to the black-box constraint of the total energy consumption of all the agents below some threshold. At the same time, the operator needs to keep the prices for different agents the same or at least no significant differences~(affine constraint) due to fairness reasons. The experiments in this paper aim to separately demonstrate the effectiveness of our algorithm in dealing with these two types of constraints. More realistic experiments with both types of constraints are left as future works.
> > I think the example given at the top of page 5 can be further clarified.
>
> Thanks for the suggestion. We have added more clarifications to this example and updated the paper accordingly.
> > Theorem 1: I think some discussions regarding the differences between the two cases would be helpful. Regarding the dependency on $\sum_{i=1}^N\sum_{j=1}^m \gamma_{i,j}^T$, if we substitute in the rate of $\gamma$ for either the SE kernel and Matern kernel, I guess this term would depend linearly on $N$ and $m$? If this is the case, then for case 1 of Theorem 1, the first term in the regret bound would have a dependency of $N^2$ and $m$. Also, it is mentioned in the paragraph below Theorem 1 that "we can trade violation for smaller regret", I think some elaborations would be helpful.
>
> Yes. And thanks for the suggestion. We have revised the manuscript accordingly by appending more discussions in the paragraph that follows the Thm. 1.
>
> > Remark 4: if I understand correctly, the assumption $\lim_{T\to\infty}\sum_{i=1}^N\sum_{j=1}^m\gamma_{i,j}^T/\sqrt{T}=0$ holds for the Matern kernel only when the function is smooth?
>
> Yes. It indeed requires the Matern kernel to have a certain level of smoothness. Actually, to satisfy this constraint, the smoothness parameter $\nu_{i,j}$ needs to be at least $\frac{d_{i,j}}{2}$, where $d_{i,j}$ is the input dimension of the corresponding black-box function. We have revised Remark 4 accordingly in the manuscript.
>
> > Figure 1: I agree that the strong violation of the proposed DMABO grows slower and slower yet the baseline of DCEI suffers from linear growth, but in this figure, the strong violations of DMABO is always larger than DCEI. It seems the relative magnitude is only reversed when the number of steps is large enough.
>
> Yes. Our theoretical results hold asymptotically. It is possible that in the initial steps, our method has a larger violation.

---

> > ### Comment · Reviewer_t8mC · 2023-12-01
> > **Thanks for the response**
> >
> > Thanks for the response and the paper revision, which clarified some of my questions. After reading the response and the other reviews, I'd like to keep my rating.

---

### Author Response · Authors · 2023-11-17
**Response to all the reviewers**

We would like to thank all the anonymous reviewers for your helpful suggestions and valuable comments. In our rebuttal, we addressed all the concerns of the reviewers. We present one-to-one responses to individual comments in the corresponding blocks. We are more than happy to address your further comments.

---

### Author Response · Authors · 2023-11-17
**Revised paper uploaded**

We have also uploaded the revised paper, with the revisions highlighted in color blue.

---

### Author Response · Authors · 2023-11-20
**Any follow-up questions/comments?**

Dear reviewers,

As the rebuttal deadline approaches within the next two days, we wanted to check in and see if you may have additional questions or comments. We are more than happy to provide further clarification on our paper, particularly concerning its contributions.

Best regards,

All the authors.

---

### Meta-Review · Area_Chair_ZNWH · 2023-12-09

**Metareview:**

The paper presents a novel algorithm for distributed multi-agent Bayesian optimization, effectively extending the primal-dual decomposition approach from single-agent to multi-agent scenarios. This extension is notable for incorporating both black-box and affine constraints within a multi-agent framework, a novel setting in constrained BO. The authors have provided a comprehensive theoretical analysis, demonstrating the algorithm's robustness in terms of regret and constraint violations. However, the reception of the paper among reviewers was mixed, primarily due to concerns regarding its incremental nature and limited experimental scope.

**Justification For Why Not Higher Score:**

The paper does not receive a higher score primarily due to its perceived incremental nature over existing algorithms and limited experimental validation (i.e. predominantly focusing on scenarios involving known affine constraints).

After the discussion phase, reviewers have acknowledged the paper's innovation in adapting and extending existing methodologies to a more complex multi-agent setting. However, post-rebuttal reviews also highlighted that while the approach is novel and the theoretical analysis is sound, the contributions appear as incremental advancements built upon established single-agent models. Such perception of the paper has tempered enthusiasm for a higher score.

**Justification For Why Not Lower Score:**

The paper has made novel contributions to distributed multi-agent Bayesian optimization. The authors have been commendably responsive to the reviewers' comments, effectively addressing concerns and providing additional clarifications that distinguish their work from prior studies (which has been beneficial in highlighting the paper's strengths, particularly its relevance to practical, real-world scenarios and its potential for future research and application).

---

### Decision · Program_Chairs · 2024-01-16

Reject